# In silico exploration of *Serratia* sp. BRL41 genome for detecting prodigiosin Biosynthetic Gene Cluster (BGC) and in vitro antimicrobial activity assessment of secreted prodigiosin

**Farhana Boby**[1]*, **Md. Nurul Huda Bhuiyan**[1], **Barun Kanti Saha**[1‡], **Subarna Sandhani Dey**[1], **Anik Kumar Saha**[1☯], **Md Jahidul Islam**[1☯], **Mahci Al Bashera**[1☯], **Shyama Prosad Moulick**[2☯], **Farhana Jahan**[1☯], **Md. Asad Uz Zaman**[3], **Sanjana Fatema Chowdhury**[2☯], **Showti Raheel Naser**[2☯], **Md. Salim Khan**[2‡], **Md. Murshed Hasan Sarkar**[2]*

**1** BCSIR Rajshahi Laboratories, Bangladesh Council of Scientific and Industrial Research (BCSIR), Dhaka, Bangladesh, **2** BCSIR Laboratories, Dhaka, Bangladesh Council of Scientific and Industrial Research (BCSIR), Dhaka, Bangladesh, **3** Department of Microbiology, Primeasia University, Dhaka, Bangladesh

☯ These authors contributed equally to this work.
‡ BKS and MSK also contributed equally to this work.
* farhana_boby@bcsir.gov.bd (FB); murshed_mbdu@yahoo.com (MMHS)

**Data Availability Statement:** All relevant data are within the paper.

## Abstract

The raising concern of drug resistance, having substantial impacts on public health, has instigated the search of new natural compounds with substantial medicinal activity. In order to find out a natural solution, the current study has utilized prodigiosin, a linear tripyrrole red pigment, as an active ingredient to control bacterial proliferation and prevent cellular oxidation caused by ROS (Reactive Oxygen Species). A prodigiosin-producing bacterium BRL41 was isolated from the ancient Barhind soil of BCSIR Rajshahi Laboratories, Bangladesh, and its morphological and biochemical characteristics were investigated. Whole genome sequencing data of the isolate revealed its identity as *Serratia* sp. and conferred the presence of prodigiosin gene cluster in the bacterial genome. "Prodigiosin NRPS", among the 10 analyzed gene clusters, showed 100% similarity with query sequences where pigC, pigH, pigI, and pigJ were identified as fundamental genes for prodigiosin biosynthesis. Some other prominent clusters for synthesis of ririwpeptides, yersinopine, trichrysobactin were also found in the chromosome of BRL41, whilst the rest displayed less similarity with query sequences. Except some first-generation beta-lactam resistance genes, no virulence and resistance genes were found in the genome of BRL41. Structural illumination of the extracted red pigment by spectrophotometric scanning, Thin-Layer Chromatography (TLC), Fourier Transform Infrared Spectroscopy (FTIR), and change of color at different pH solutions verified the identity of the isolated compound as prodigiosin. *Serratia* sp. BRL41 attained its maximum productivity 564.74 units/cell at temperature 30°C and pH 7.5 in two-fold diluted nutrient broth medium. The compound exhibited promising antibacterial activity against Gram-positive and Gram-negative bacteria with MIC (Minimum Inhibitory

**Funding:** All financial support for this work was provided by Bangladesh Council of Scientific and Industrial Research (BCSIR), Dhaka, Bangladesh and it has no role in study design, data collection and analysis, decision to publish or preparation of manuscript.

**Competing interests:** All authors have declared that they have no competing interests.

Concentration) and MBC (Minimum Bactericidal Concentration) values ranged from 3.9 to15.62 µg/mL and 7.81 to 31.25 µg/mL respectively. At concentration 500 µg/mL, except in *Salmonella enterica* ATCC-10708, prodigiosin significantly diminished biofilm formed by *Listeria monocytogens* ATCC-3193, *Pseudomonas aeruginosa* ATCC-9027, *Escherichia coli (*environmental isolate*), Staphylococcus aureus (*environmental isolate). Cellular glutathione level (GSH) was elevated upon application of 250 and 500 µg/mL pigment where 125 µg/mL failed to show any free radical scavenging activity. Additionally, release of cellular components in growth media of both Gram-positive and Gram-negative bacteria were facilitated by the extract that might be associated with cell membrane destabilization. Therefore, the overall findings of antimicrobial, antibiofilm and antioxidant activities suggest that in time to come prodigiosin might be a potential natural source to treat various diseases and infections.

## Introduction

In recent years, the emergence and rapid spread of antibiotic resistance has become a matter of concern for the world health community. Therefore, scientists are continuously searching for potential natural sources to combat these problems and microbial products in this regard are considered as a good choice over plants and animals. Thermal stability, ease of cultivation, genetic manipulation, and availability throughout the year are some of the fascinating features that have drawn the attention of scientists to work with bio-products [1]. In these circumstances, prodigiosin, a secondary metabolite of bacteria is an important candidate to be used as a potential alternative to antibiotics. It shows a broad spectrum of antibacterial activity against Gram-positive and Gram-negative bacteria along with anti-cancer, anti-fungal, anti-protozoal, anti-malarial, anti-larval, and immunosuppressive activity [2, 3]. The significant antioxidant activity of prodigiosin indicates its competency as a colorant and therefore, can be used in the food and cosmetics industries [4]. It is reported that 99% oxidation can be inhibited by prodigiosin at a concentration of 10 µg/mL [5]. Prodigiosin is a tripyrrole red pigment synthesized by prodigiosin biosynthesizing gene cluster (pig cluster) as two key intermediates: 2-methyl-3-n-amylpyrrole (MAP) and 4-methoxy-2, 2'- bipyrrole–5-carbaldehyde (MBC), and form prodigiosin through a condensation reaction [6, 7]. Among 6 members of prodiginine family (prodigiosin, undecylprodigiosin, cycloprodigiosin, metacycloprodigiosin, prodigiosin R1 and streptorubin B), prodigiosin is the most influential one with more therapeutic values [8].

Several bacterial and fungal species like *Serratia marcescens*, *S. rubidaea*, *Pseudovibrio detitrificans*, *Pseudoalteromonas rubra*, *Vibrio psychroerythrous*, *V. gazogens*, *Streptomyces lividans*, *Nocardia* spp. etc. can synthesize prodigiosin as secondary metabolite [9]. *Serratia marcescens* was the bacterium from which prodigiosin was first extracted though the role of prodigiosin in *Serratia marcescens* is still unrevealed [8]. Possibly this may be produced by host bacteria during interspecies competition in a nutrient-limiting environment where prodigiosin inhibits the growth of other Gram-positive and Gram-negative bacterial species persisting in that environment [8, 10]. It is a Gram-negative bacterium that belongs to the family of Enterobacteriaceae [11]. *Serratia* sp. is ubiquitous in nature and is mostly found in water, soil, plants, and air [12]. Though the both pigmented and non-pigmented variants of *Serratia marcescens* are pathogenic for human, researchers claim that the non-pigmented strains are more virulent due to cytotoxin production and antibiotic resistance [12]. Commonly used media for prodigiosin biosynthesis by *Serratia marcescens* include nutrient broth, peptone, glycerol broth, optimized production media, etc. where nutrient broth exhibits a higher yield of prodigiosin at 28–30°C

compared to others [13]. Physicochemical parameters like temperature, pH, incubation time, and quorum sensing play a very influential role in prodigiosin biosynthesis by *Serratia marcescens* [13]. Mechanisms work behind antimicrobial activity of prodigiosin include deoxyribonucleic acid (DNA) cleavage, modulation of pH, and cell cycle inhibition [9]. Formation of Reactive oxygen species (ROS) [14], and photo cytotoxicity can also be two other possible mechanisms of antimicrobial activity [15].

Prodigiosin has been found to inhibit biofilm formation by pathogenic bacteria like methicillin-resistant *Staphylococcus aureus* (MRSA), *Staphylococcus aureus*, *Enterococcus faecalis*, *E. coli*, *Salmonella typhimurium*, *Pseudomonas aeruginosa* though a finding showed substantial biofilm acceleration activity of prodigiosin in *Staphylococcus aureus* and *Pseudomonas aeruginosa* [16, 17]. Despite being toxic to bacterial and cancer cells, it shows little or no cytotoxicity to normal cells [18, 19]. These characteristic features of prodigiosin provides evidence to work with this metabolite as an alternative to drugs.

So, in our study we have addressed the bacteriostatic/bactericidal behavior of prodigiosin towards both Gram-positive and Gram-negative bacterial pathogens. *Serratia* sp. BRL41 is the first reported prodigiosin-producing bacteria isolated from Barhind soil in northern region of Bangladesh. Prior checking antimicrobial activity, prodigiosin was first identified and then characterized. Another aspect of this study was to utilize the whole genome sequence data of the isolate to identify the Biosynthetic Gene Clusters (BGC) associated with secondary metabolite biosynthesis to facilitate the understanding of prodigiosin biosynthesis and its regulation mechanism with the identification of core biosynthetic gene in BRL41. To manifest its competency as industrial stain the study explored antibiotic resistance and virulence genes in the chromosome of the isolate and accomplished with a positive result. We have investigated the membrane integrity of Gram-positive and Gram-negative bacterial cell membrane upon treatment with prodigiosin and for the very first time its role as a catalyst to elevate glutathione level (GSH) into human blood cell. These findings of the biological and pharmaceutical activities of prodigiosin provide evidence of its potency as an active molecule that is likely to be used for pharmaceutical formulations.

## Materials and methods

### Source of bacteria and culture media

All American Type Culture Collection (ATCC) were obtained from Fruits and Food Research Division BCSIR Laboratories, Rajshahi, Bangladesh. Only *Escherichia coli* and *Staphylococcus aureus* were isolated from the local soil sample. *Serratia sp*. BRL41 was screened out from the garden soil of BCSIR Laboratories, Rajshahi. The sample was collected aseptically in a sterile, clean, dry sample bag with the help of a spatula and stored in the refrigerator at 4˚C until further processing. Short term-storage of cultures were maintained on agar slant at 4˚C and for long-term storage cultures were kept on 50% glycerol (V/V) at -20˚C. All bacteria were cultured on nutrient agar (Himedia, India) at 37˚C except BRL41which was allowed to propagate at 30˚C.

### Isolation and identification of bacteria

In search of a pigment producing isolate, BRL41 was screened out from the mixed microbiota of soil through a sequential screening at varying temperature and incubation time that resulted in appearance of few colored colonies on nutrient agar medium. Some of the pigmented colonies were picked up and sub-cultured several times to obtain pure culture. From there, we dealt with only red pigmented one and the morphological trait was determined by microscopic observation and biochemical analysis in a semi-automated micro station ID system using the

software program Microlog[TM] 6.2 according to the manufacturer's instructions. For further confirmation whole genome sequencing was performed using Illumina MiniSeq platform.

## Genome sequence assembly, annotation and phylogenetic and Biosynthetic Gene Cluster (BGC) analysis

DNA from the isolate was extracted using Wizard® Genomic DNA Purification Kit (Promega, catalog no. PR-A1120) and whole genome sequencing was done by Illumina MiniSeq platform. Library preparation was conducted as per the manufacturer's protocol. After obtaining the raw data, the assembly of the genome was performed using the Shovill pipeline [20] that utilized SPAdes as its core and is specific for bacterial genomes from Illumina paired end reads. Raw sequencing data generated from this study was deposited at the NCBI sequence read archive (SRA) under accession Bio project PRJNA998550. Phylogenetic analysis was performed by Type (Strain) Genome Server (TYGS) using maximum-likelihood method [21] and the phylogenetic tree was visualized using iTOL [22]. For further confirmation of the similarity of the isolate, Average Nucleotide Identity (ANI) was calculated using FastANI [23]. Biosynthetic gene clusters responsible for the secondary metabolite production were identified using antiSMASH [24]. Moreover, antibiotic resistant genes and virulence genes were profiled by CARD browse and VFDB analyzer [25]. This identified bacterial isolate was used to conduct further studies.

## Production of pigment by BRL41

To facilitate pigment production from isolated bacteria, nutrient broth was inoculated with the respective bacteria and incubated at 30°C for 96 hours. Then the culture was centrifuged at 6000 rpm for 10 min and cell pellet was suspended in methanol and sonicated for 1 hour followed by further centrifugation. Pigment production per unit of the bacterial cell was estimated using the following formula [15]

$$\textbf{Prodigiosin (unit per cell)} = \frac{[OD_{499} - (1.38 \times OD_{620})] \times 1000}{OD_{620}}$$

Here, $OD_{499}$ is the pigment absorbance in culture and $OD_{620}$ is the bacterial cell absorbance. $(1.38 \times OD_{620})$ stands for the number of bacterial cells in culture, where the value of $OD_{499}$-$(1.38 \times OD_{620})$ represents the absorption of pigment. To express prodigiosin units on a per cell basis, the absorption value $OD_{499}$-$(1.38 \times OD_{620})$ was divided with $OD_{620}$ and to avoid working with small numbers ($<1$) a factor of 1000 was included here,

## Identification of pigment

Primarily, identification was done by the characteristic changes of pigment color in acidic and basic solvent. For this, one fraction of the extracted color was acidified by 1N hydrochloric acid (HCl) and the other fraction was alkalified by 1N sodium hydroxide (NaOH) that caused a visual change of pigment's color. Further prodigiosin was identified by Thin Layer Chromatography (TLC) on a silica gel plate where 1 drop of pigment was placed at the bottom and allowed to move upward using ethyl acetate as the mobile solvent. From TLC, Rf value was calculated using the following equation:

$$R_f = \frac{\text{Distance traveled by molecule}}{\text{Distance traveled by mobile phase}}$$

Maximum absorption wavelength ($\lambda_{max}$) was determined by UV-visible spectral scan between 200 to 900 nm in Hitachi U-2910 UV spectrophotometer using Dimethyl sulfoxide (DMSO) as blank. Fourier Transform Infrared Spectroscopy (FTIR) (PerkinElmer Spectrum IR Version 10.6.2) demonstrated the molecular skeleton structure of prodigiosin where sample was mixed with methanol and analyzed within the range of 4000 to 500 cm$^{-1}$.

## Optimization of pigment production conditions

To find out optimum conditions for maximum pigment production, some important parameters like incubation time (24, 48, 72, 96 hour), temperature (25, 28, 30, 37° C), pH (5.00 to 8.5) were checked using nutrient broth as basal media. After each treatment color intensity was measured spectrophotometrically at 537 nm where the highest absorbance signified the highest pigment production on that particular condition.

## Selection of extraction solvent

Different organic solvents like n-hexane, methanol, ethanol, DMSO, Acetonitrile: n-hexane (4:6), and 0.5% acidified methanol were utilized to select an appropriate extractor of cellular pigment. For this, 5 mL of 96 hours culture of BRL41 was collected and centrifuged to discard supernatant. In remaining cell pellet, 5 mL of the above-mentioned solvents were added, and again centrifuged after 1 hour. Absorbance of collected supernatant was taken in UV spectrophotometer where higher absorbance resulted from higher pigment extraction. The dry powder of pigment was derived from solvent evaporation using rotary evaporator at 65°C followed by air drying for few hours and then stored in cool and dry place.

## Antimicrobial assay of prodigiosin

The antimicrobial activity of prodigiosin was appraised using the standard agar well diffusion method on Mueller-Hinton agar medium [26]. A standard chloramphenicol solution (30 μg/mL) was used as a positive control and 10% DMSO, where prodigiosin was suspended, was used as the negative control. Antimicrobial activity of this metabolite was tested against five pathogens (*Salmonella enterica* ATCC-10708, *Listeria monocytogenes* ATCC-3193, *E. coli*, *staphylococcus aureus*, *Pseudomonas aeruginosa* ATCC 9027) at different prodigiosin concentrations (500 μg/mL, 250 μg/mL, and 125 μg/mL). At first, the optical density of freshly cultured bacteria was adjusted between 0.4 to 0.6 by using fresh nutrient broth solution so that the cell density could be approximately $1 \times 10^8$ CFU/mL. 100μL of the culture was then spread on agar medium and allowed to dry for 5 minutes. 30 μL of each dilution of prodigiosin was loaded in 6 mm agar well along with positive and negative control. The plates were then incubated at 37°C overnight and the zone of inhibition surrounding the well was measured and recorded. To reproduce this result, each experiment was done in triplicate sets.

## Determination of Minimum Inhibitory Concentration (MIC) and Minimum Bactericidal Concentration (MBC)

MIC of prodigiosin was assessed against these five species of bacteria by macro dilution method in test tubes [27]. Briefly, a solution of prodigiosin with a final concentration of 500 μg/mL was prepared using 10% DMSO and 1ml of preparation was added in the 1st tube of density adjusted culture of each bacterium. It was then serially diluted to obtain a set of culture tube having prodigiosin concentrations from 500 μg/mL to 0.24 μg/mL. Tube containing culture broth and inoculum represented a positive control whereas a tube with culture broth, inoculum, and antibiotic chloramphenicol was used as a negative control. After overnight

incubation, the growth pattern of bacteria was observed to determine the minimum inhibitory concentration based on the absence of visible growth. MBC was determined by spreading 10 μL of culture from each tube on nutrient agar plate to find out the concentration that caused the complete death of bacterial cells. Concentration for which no bacterial growth was observed on the plate was considered as MBC.

### Anti-biofilm activity of prodigiosin

The worth of prodigiosin in hindrance of bacterial surface adhesion, the primary step of biofilm formation, was apprised by crystal violet assay [28]. For this, a serial dilution of prodigiosin was prepared in a microtiter plate and bacterial suspension was added to each well to reach a final volume of 100 μL. Herein, nutrient broth represented blank whereas a mixture of nutrient broth, DMSO, and 10 μL of bacteria served as control. After 48 hours the plate was washed twice with 1×PBS solution followed by the addition of 200 μL of 99% methanol. Soon after 20 minutes, it was drained out and allowed to dry for next 15 minutes. To stain bacterial cell, 200 μL of 0.1% crystal violet was added to each well, waited for 5 minutes and dried the plate off by unloading the dye. Finally, to solubilize the crystal violet attached to the bacterial cell, 95% ethanol was added and mixed and the optical density was measured at 450 nm using a microtiter plate reader. Biofilm inhibition % was calculated using the following formula [28]

$$\frac{(\text{OD}_{\text{control}} - \text{OD}_{\text{treated}})}{\text{OD}_{\text{control}}} \times 100$$

### Antioxidant assay by estimating reduced glutathione (GSH) content in cell

Accomplishment of GSH measurement followed the methodology described by Bomzon et al., 2010 [29] with slight modification. Shortly, prodigiosin of varying concentrations (125, 250, 500 μg/ml) were taken in three separate tubes and erythrocyte suspension was added and mixed by gentle inversion. Erythrocytes with PBS was served as control and all the tubes were incubated at 37˚C for 24 hours. After centrifugation of reaction mixtures, water addition facilitated cell lysis followed by addition of metaphosphoric acid that resulted in precipitation of cell lysate. Immediately after centrifugation both control and treatment tubes were treated by 0.01 M sodium phosphate buffer (pH 7.5), 5 mM EDTA, 10 mM 5, 5-dithio-bio-nitrobenzoic acid (DTNB), 2mM NADPH, 100 U/ml glutathione reductase to reach a final volume 1ml. Then absorbance was taken at 412 nm against blank.

### Protein release assay -Bradford method

This assay served as an indirect method to assess bacterial membrane impairment by measuring the concentration of protein in the supernatant of bacterial growth medium. Here, *E. coil* and *Listeria monocytogens* ATCC- 3193 were used as representative of Gram-negative and Gram-positive bacteria respectively. Each bacterial suspension was treated with 500 μg/mL prodigiosin for 1, 2 and 3 hours. Following centrifugation at 300×g for 30 min, the supernatant from each tube was used to estimate protein by the Bradford protein estimation method using bovine serum albumin (BSA) as standard [30]. Reproducibility of the result was gained by performing each experiment in triplicate sets.

### Statistical analysis

All statistical analysis was performed using IBM SPSSS statistics 22 software. Data were expressed as mean ± standard deviation. One-way ANOVA followed by Tukey test was

conducted for comparing means of zone of inhibition. p value < 0.05 were considered to be statistically significant at 95% confidence level.

## Result

### Identification of *Serratia marcescens* BRL41

Microscopic observation revealed the isolated bacteria as Gram-negative, rod-shaped, and motile that formed moderate round-shaped red colonies on nutrient agar medium. Based on the utilization of different metabolic substrates and the ability to grow in the presence of certain inhibitory chemicals, the result from MicroStation GEN III database 2.8.0 library showed 88% similarity of this strain with *Serratia marcescens*. Genome sequencing data depicted the highest sequence similarities with *Serratia marcescens* that was further supported by ANI results, showing 98.15% nucleotide similarity with the reference genome of *Serratia marcescens* (GCF_003516165.1). Sequence alignment and construction of the phylogenetic tree revealed that the isolate is very closely related to strain *Serratia marcescens* subsp., *sakeunsis* KCTC 42172, and *Serratia marcescens* ATCC 13880 "**Fig 1**".

### Analysis of Biosynthetic Gene Cluster (BGC)

AntiSMASH analysis for the secondary metabolite producing BGC visualized the presence of 10 different biosynthetic gene clusters in BRL41 chromosome where the hybrid cluster "Prodigiosin NRPS" showed 100% similarity with the query sequence (**Table 1**). The core prodigiosin biosynthetic genes pigC, pigH, pigI, and pigJ co-existed with five additional biosynthetic genes and all of them were regulated by cueR "**Fig 2**". Similarly, two other clusters, supposed to produce ririwpeptide and yersinopine, exhibited 100% similarity with query sequences in BGC analysis.

Tree scale : 0.01

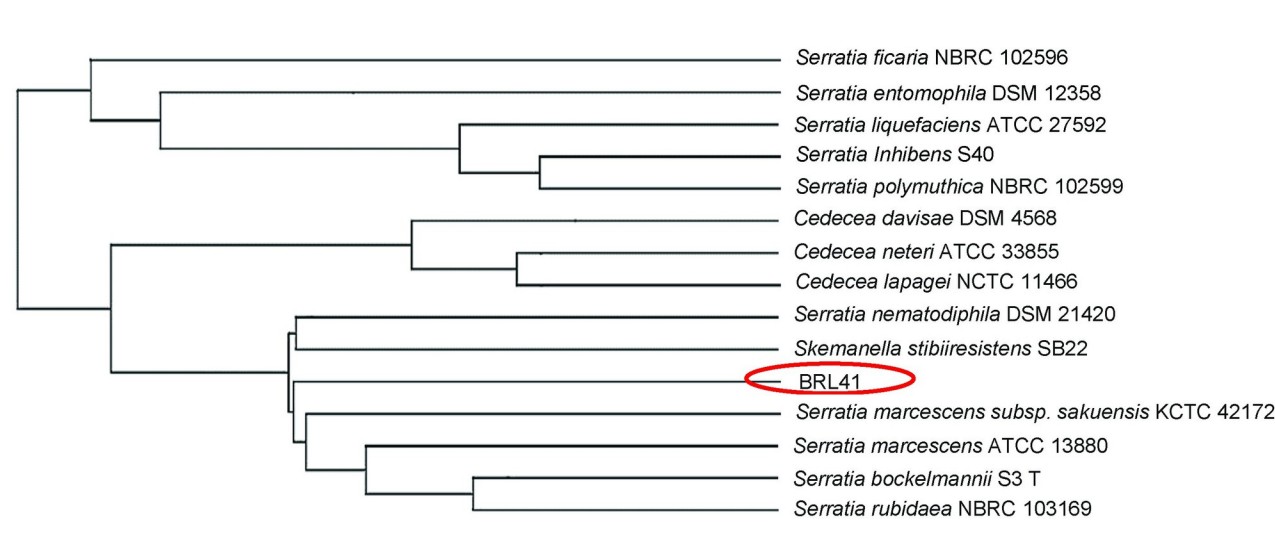

**Fig 1. Phylogenetic position of *Serratia* isolate BRL41 in relation to its closest strain.**

**Table 1. Biosynthetic gene clusters for secondary metabolite producing genes in BRL41 strain using AntiSMASH.**

| BGC Type | From | To | Most similar known cluster | Similarity |
|---|---|---|---|---|
| prodigiosin, NRPS | 446,225 | 507,414 | Prodigiosin | 100% |
| NRPS | 132,409 | 176,344 | ririwpeptide A/ririwpeptide B/ririwpeptide C | 100% |
| opine-like-metallophore | 121,810 | 143,907 | Yersinopine | 100% |
| NRP-metallophore, NRPS | 304,172 | 354,869 | trichrysobactin/cyclic trichrysobactin/chrysobactin/dichrysobactin | 46% |
| NRPS | 1,076,754 | 1,125,264 | trichrysobactin/cyclic trichrysobactin/chrysobactin/dichrysobactin | 38% |
| NRPS | 382,689 | 459,204 | 5-dimethylallylindole-3-acetonitrile | 33% |
| NRPS | 7,283 | 45,973 | microcin E492 | 18% |
| Thiopeptide | 136,026 | 162,470 | O-antigen | 14% |
| RRE-containing | 1 | 1,641 | lankacidin C | 13% |
| Betalactone | 119,114 | 144,783 | | |

## Presence of antibiotic resistance and virulence genes

In "perfect" algorithm setting of CARD database, no resistance gene was found in the genome of the isolate. On the other hand, with very low percentage of identity, some resistance genes were found in "Strict" setting but they were against commonly available beta-lactam, first-generation antibiotics. Though the VFDB analysis showed 25 virulence genes in the chromosome, they were not pathogenic genes at all cause being a gram-negative bacterium, having some structural endotoxins in *Serratia* is quite normal.

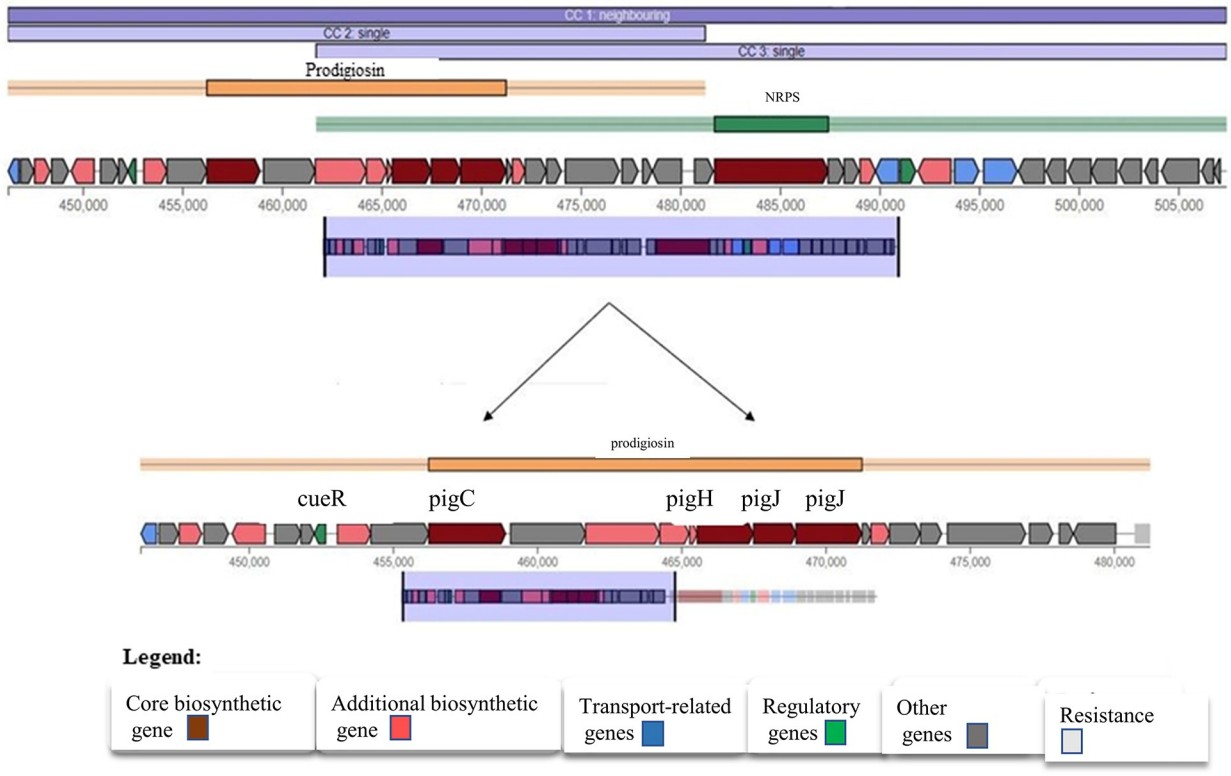

**Fig 2. Prodigiosin gene cluster showing gene pool involve in prodigiosin production.**

## Production and characterization of pigment

Production of prodigiosin was carried out in 2-fold diluted nutrient broth at ambient pH (7.5) and temperature (30°C). Pigment isolated from BRL41 exhibited brown and dark pink color in basic and acidic solutions (pH 10 and pH 2 respectively), an evidence supported being prodigiosin of secreted the pigment. Upon subjected to TLC using ethyl acetate as the mobile phase, it displayed an $R_f$ value of 0.87 "**Fig 3**".

UV absorbance profile scanned from 200–900 nm presented a sharp spectral peak at 537 nm which is a vital characteristic of prodigiosin "**Fig 4**". In FTIR spectral analysis, influential peaks were observed at 2925.40 (aromatic CH) and 1625.43 (aromatic C = C) cm$^{-1}$ while the other well-built peaks were at 3268.17, 2854.23, 1722.06, 1404.78, 1150.16, 1075.09, 1016.18 cm$^{-1}$ "**Fig 5**".

## Different physicochemical factors controlled the production and extraction of prodigiosin

Different physicochemical factors were found to affect prodigiosin production by *Serratia marcescens* BRL41. Commencing at 25°C, synthesis reached to its maxima at 30°C with no pigment production at 37°C "**Fig 6A**". The highest pigment production (564.74 units /cell) was found at pH 7.5 and after that it started to cease "**Fig 6B**". Under ambient conditions, an elevated pigment production was noticed after 72 hours of incubation that soared at its peak when reached to 96 hours "**Fig 6C**". Nevertheless, ethanol and DMSO showed almost the same efficacy, acidification of methanol using 0.5% HCl facilitated an escalation in pigment liberation "**Fig 6D**".

## Prodigiosin exhibited antimicrobial activity against pathogenic bacteria

Five bacterial pathogens were utilized to manifest the inhibitory effect of prodigiosin on bacterial proliferation by agar diffusion method according to Clinical Laboratory Standards. Although prodigiosin significantly inhibited the growth of the tested pathogens at concentration 500 μg/mL and 250 μg/mL "**Fig 7**, **Table 2**", it failed to affect the growth of *E. coli*, *Pseudomonas aeruginosa* ATCC-9027, *and Salmonella enterica* ATCC-10708 at 125 μg/mL. MIC and MBC values of crude extract were calculated for each of these 5 pathogens "**Table 3**" where a higher MIC (> 10 μg/mL) was required to hinder the growth of *E. coli*, *Staphylococcus aureus*, and *Listeria monocytogenes* ATCC-3193 compared to *Pseudomonas aeruginosa* ATCC-9027, and *Salmonella enterica* ATCC-10708. For all pathogens except *Pseudomonas aeruginosa* ATCC-9027, high concentrations of crude extract (> 10 μg/mL) were indispensable to attain bactericidal activity.

## Inhibition of biofilm formation

Biofilm formation by pathogenic bacteria facilitates their irreversible attachment to the growing surface together that results in antibiotic resistance development concerning virulent gene transmission and growth enhancement. Adhesion to surface is the initial stage of biofilm formation and therefore, inhibition of adhesion to surface prevent bacteria from biofilm formation. Taking it into consideration, the anti-biofilm activity was assessed against these five pathogens. A significant reduction of surface adhesion of *Listeria monocytogenes* ATCC-3193 (≥50%) and *Pseudomonas aeruginosa* ATCC 9027 (50%) were found at a concentration of 500 μg/mL. Prodigiosin also deduced ≤50% biofilm formation by *E. coli* and *Staphylococcus aureus* while the treatment failed to change the significant amount of biofilm produced by *Salmonella enterica* ATCC-10708 (p = 0.91) "**Table 4**".

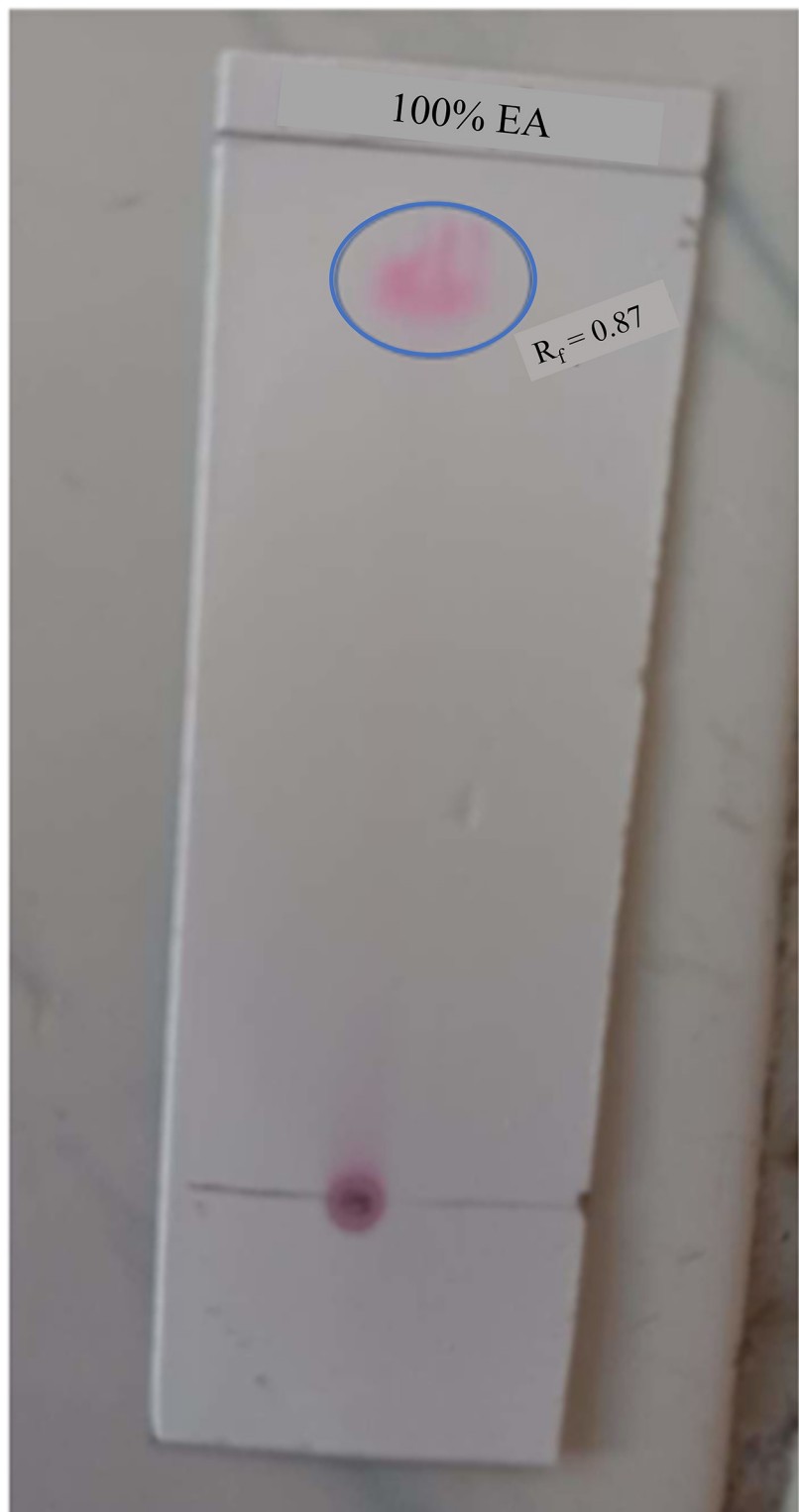

**Fig 3. Chromatographic analysis of prodigiosin by Thin Layer Chromatography (TLC) showing $R_f$ value 0.87.**

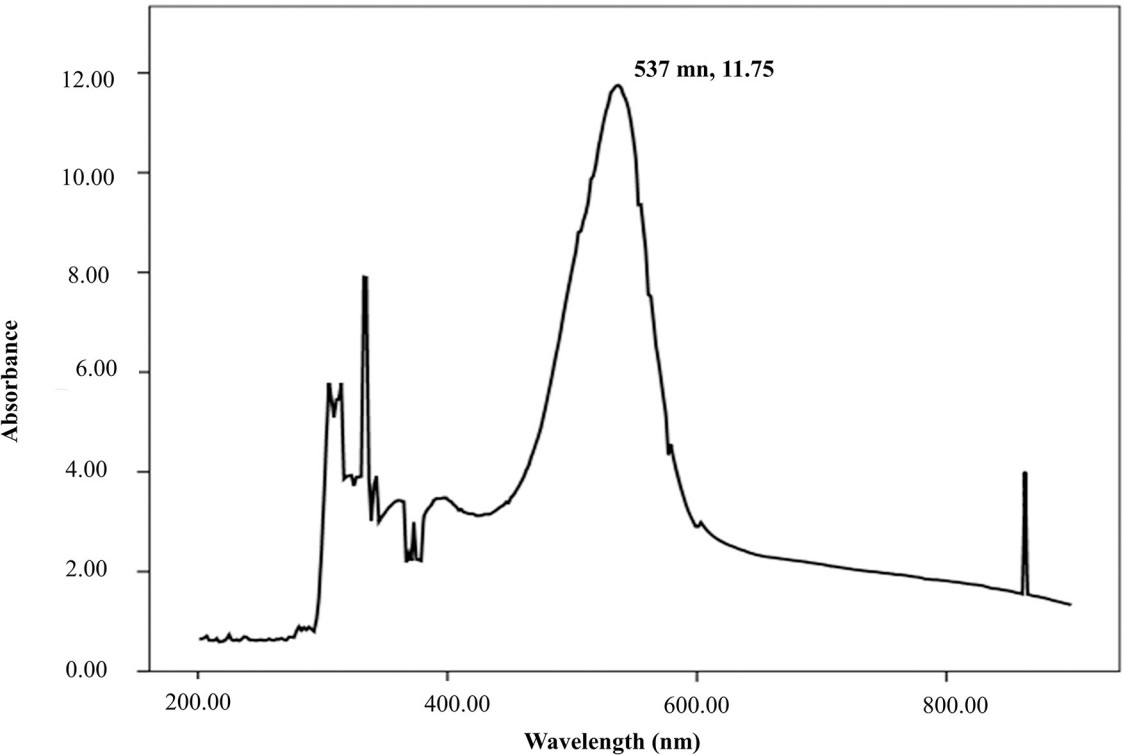

**Fig 4. UV-Vis absorption profile (200–900 nm) of crude prodigiosin with a maximum absorption at 537 nm.**

### Prodigiosin increased cellular GSH level

Being a signaling molecule, GSH is involved in antioxidant defense protecting cell from oxidative damage by Reactive Oxygen Species (ROS). Over the range of 125–500 μg/ml, prodigiosin significantly increased (p< 0.05) GSH in treated cell at concentration 250 and 500 μg/ml "**Fig 8**" where 125 μg/mL failed to stimulate GSH synthesis. This finding is an evidence towards prodigiosin's efficacy as an antioxidation stimulating agent.

### Prodigiosin affects bacterial membrane integrity

The possible damage of membrane integrity of prodigiosin treated cell was evaluated by measuring the released proteins after 1, 2, and 3 hours of treatment "**Fig 9**". After 1 hour, low level of protein detection indicated that the membrane integrity was partially affected. Protein concentration was significantly higher compared to the control group when the bacteria were exposed to the extract for 2 hours. Then surprisingly the protein concentration of both species decreased in the supernatant after 3 hours of treatment.

### Discussion

Ecological competition like antibiotic and nutrient stress as well as space limitations exert a selective pressure that force bacteria towards the synthesis of bioactive metabolites on that competitive communities. Despite a number of studies have demonstrated the potential broad-spectrum antimicrobial and antioxidant activity of prodigiosin, the precise mechanism underlying these are very poorly understood. We herein had designed an investigation focusing on pharmaceutical potencies of prodigiosin, particularly as antimicrobial and anti-oxidant

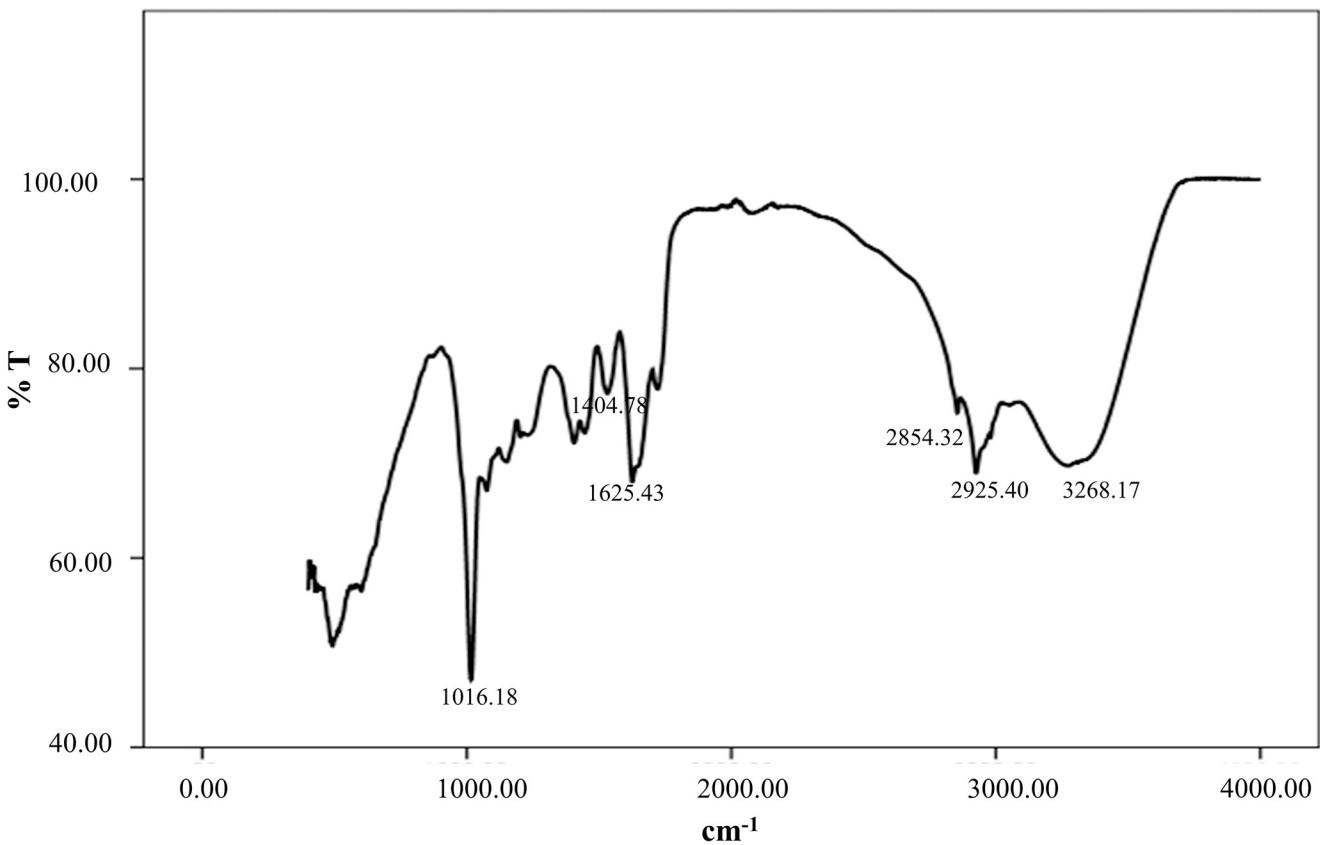

**Fig 5. FTIR spectra of crude prodigiosin extract with major stretches.**

stimulator, from a new bacterial isolate. *Serratia marcescens* BRL41 was the very first reported bacteria isolated from Pleistocene terraces in the Bengal basin of Bangladesh having prodigiosin production capabilities. Upon isolation, BRL41 was subjected to different biochemical tests, whole genome sequencing, and phylogenetic analysis to confirm its identity as *Serratia marcescens*. Along with prodigiosin, BGC analysis of genome showed the presence of other secondary metabolite synthesizing gene clusters that have opened a new door of possibilities to avail it as a potential source of some important bioproducts. Here pigC, pigH, pigI, and pigJ were detected as the prime biosynthetic genes regulated by transcription regulator cueR. As the insertion of transposons or other genomic elements on BGC can significantly alter prodigiosin production in *Serratia marcescens* [31, 32], this above-mentioned finding can be a useful basis of identifying and studying the critical points where mutations or other genomic changes can alter prodigiosin synthesis. However, existence of prodigiosin BGC has also been reported in other bacteria like *Vibrio* sp., *Pseudomonas* sp. that exhibit species and strain dependent variation in the number of Open Reading Frame (ORF) [31, 33, 34]. Absence of virulence and resistance gene in BRL41 genome is another impressive finding as it declares the fitness of this isolate to make use for commercial production of prodigiosin. The distinct color changing attributes of prodigiosin in different pH solutions has made it an important candidate to be used in food, cosmetics, and textile coloring industries. Like BRL41 such pH-dependent color change was the characteristic of prodigiosin pigment isolated from *Serratia* sp. KH-95 [35]. A single band with an $R_f$ value of 0.87 in TLC aligns with the $R_f$ value of methanol and DMSO extract of prodigiosin from *Serratia marcescens* ATCC-13880 (0.87 and 0.89 respectively) [36].

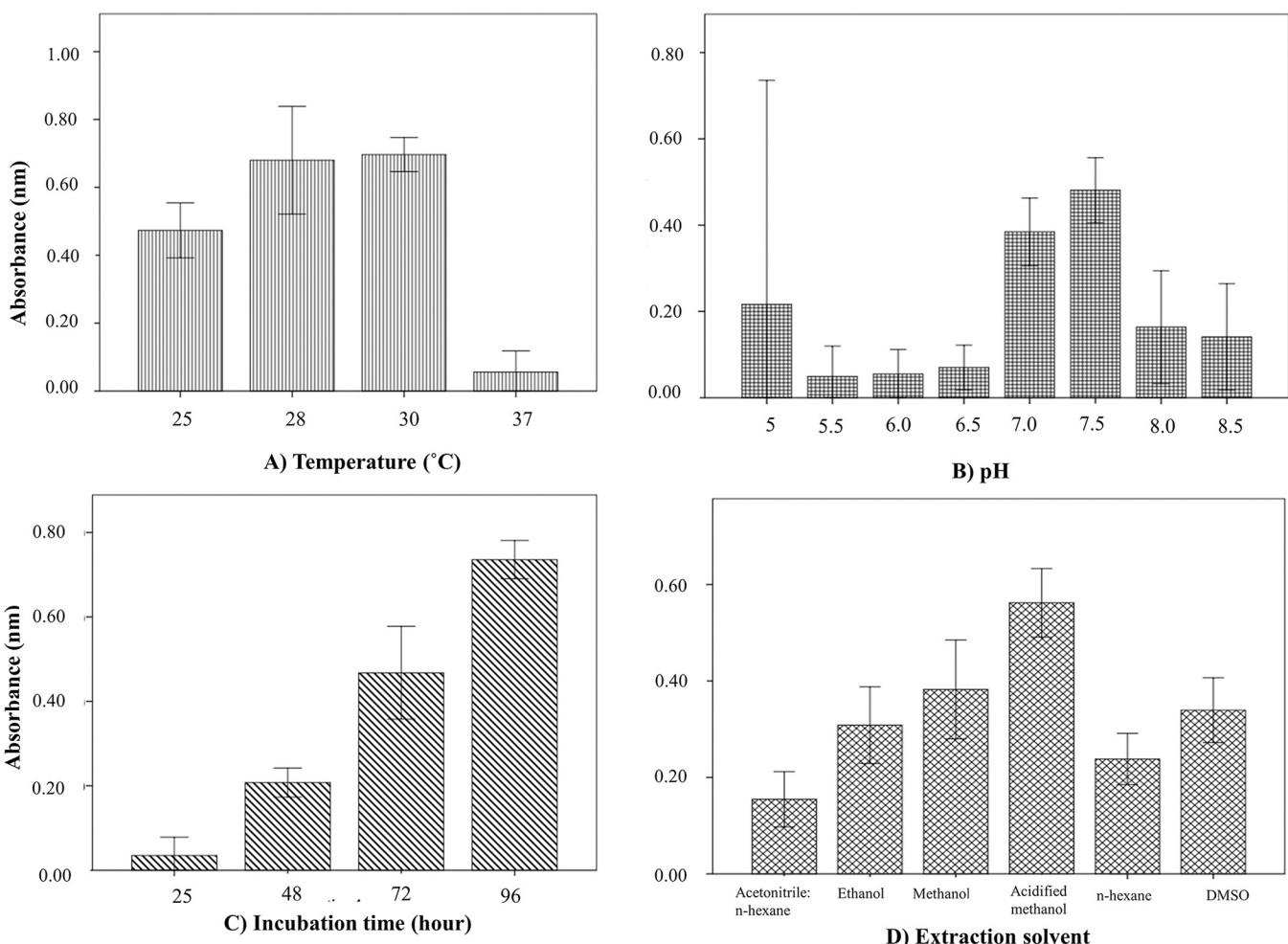

**Fig 6. Effect of different physiochemical factors in prodigiosin production and pigment extraction.** A) Temperature. B) pH. C) Incubation time. D) Solvent on prodigiosin extraction.

UV-Vis spectral analysis of prodigiosin dissolved in DMSO manifested a λ-max at 537 nm that was in line with absorption maxima at 538 nm for the methanolic extract of prodigiosin from *Serratia marcescens* [36]. This slight difference could be contributed by the variation of dissolving solvent and pH as well. The single peak at 537 nm and the absence of any other dominant

**Table 2. Zone of inhibition (mm) around the well impregnated with chloramphenicol (positive control) and prodigiosin (500 μg/ml, 250 μg/ml, 125 μg/ml).**

|  | Concentration | Tested bacteria | | | | |
|---|---|---|---|---|---|---|
|  | (μg/ml) | *E. coli* | *Staphylococcus aureus* | *Listeria monocytogens* ATCC-3193 | *Pseudomonas aeruginosa* ATCC-9027 | *Salmonella enterica* ATCC-10708 |
| Prodigiosin | 500 | 25.61 ± 0.15[a] | 18.66 ± 0.0.33[b] | 18.56 ± 0.22[b] | 24.73 ± 0.15[a] | 19.60 ± 0.05[b] |
|  | 250 | 16.89 ± 0.40[a] | 14.00 ± 0.45[b] | 13.00 ± 0.20[b] | 16.61 ± 0.11[a] | 7.77 ± 0.39[c] |
|  | 125 | No zone | 9.00 ± 0.00[a] | 9.00 ± 0.00[a] | No zone | No zone |
|  | (control) | 26.00 ± 0.58[b] | 27.67 ± 0.33[a] | 27.67 ± 0.33[a] | 26.33 ± 033[a, b] | 25.00 ± 0.00[b] |

Values are expressed as Mean ± SEM (n = 3) where SEM signifies standard error of mean. Analysis was performed with one-way ANOVA followed by Tukey Post Hoc comparisons. Mean containing different letters in same row describe significant difference of result at 5% level of Significance ($p \leq 0.05$).

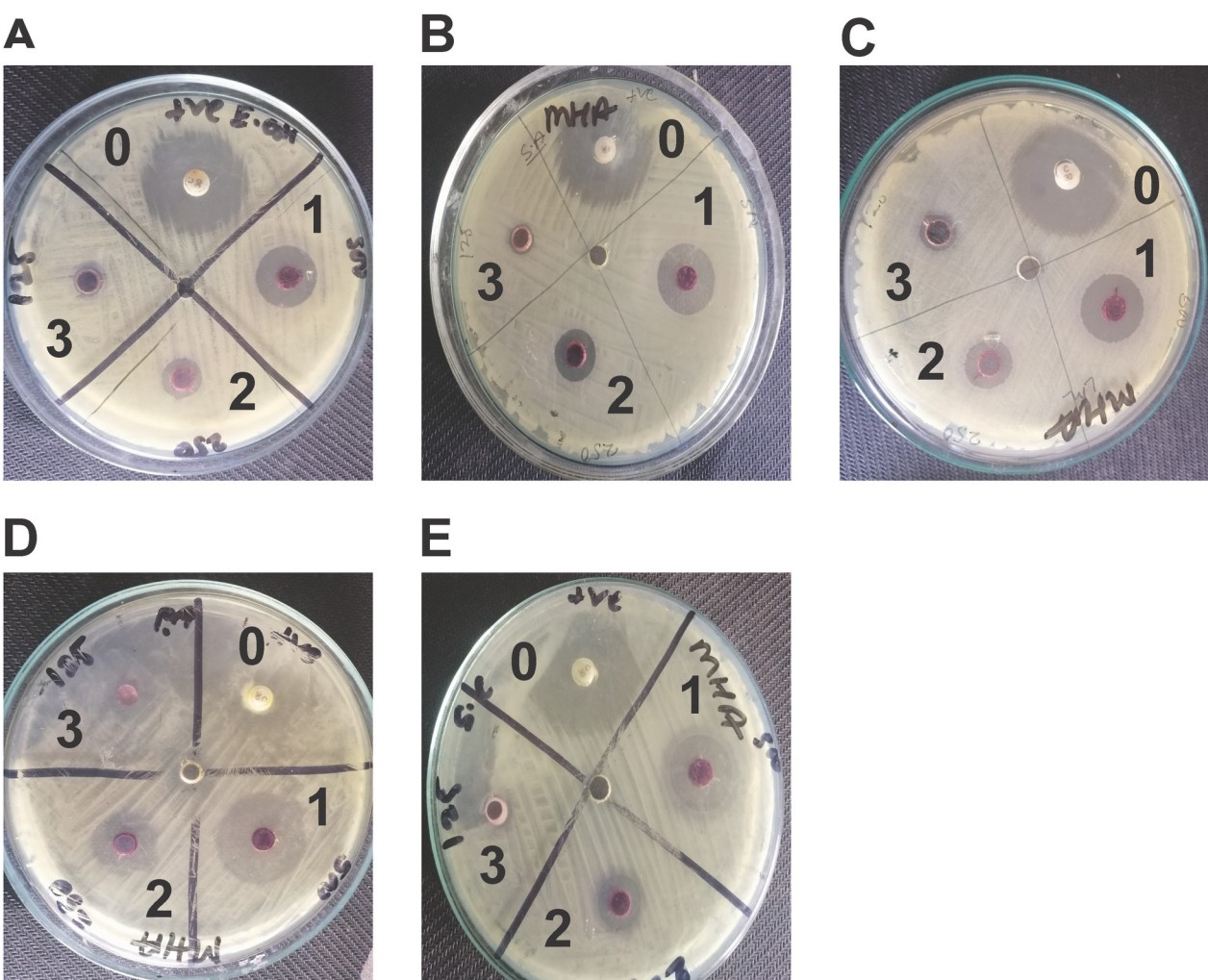

**Fig 7. Zone of inhibition surrounding wells impregnated with chloramphenicol (30 µg/mL) and prodigiosin.** A) *E. coli*. B) *Staphylococcus aureus*. C) *Listeria monocytogens* ATCC-3193. D) *Pseudomonas aeruginosa* ATCC-9027. E) *Salmonella enterica* ATCC-10708. Here, 0 = Control; 1 = 500 µg/ml; 2 = 250 µg/ml; 3 = 125 µg/ml.

peak or disturbances certified the purity of prodigiosin in the extracted sample and therefore, no further purification step was carried out. In FTIR, the most influential peaks at 2925.40 (aromatic CH), 1625.43 (aromatic C = C) cm$^{-1}$ matched with the earliest study spotted dominant bands of prodigiosin from *Serratia* sp. KH-95 at 2928 cm$^{-1}$ (aromatic CH) and 1602 cm$^{-1}$ (aromatic C = C) [35], another confirmation towards being prodigiosin of the pink extract from isolate BRL41.

**Table 3. Minimum Inhibitory Concentration (MIC) and Minimum Bactericidal Concentration (MBC) of prodigiosin for selected pathogens.**

| Bacteria | Minimum Inhibitory Concentration(µg/ml) | Minimum Bactericidal Concentration (µg/ml) |
|---|---|---|
| *E coli* | 15.62 | 31.25 |
| *Staphylococcus aureus* | 15.62 | 15.62 |
| *Listeria monocytogens* ATCC-3193 | 15.62 | 31.25 |
| *Pseudomonas aeruginosa* ATCC-9027 | 3.9 | 7.81 |
| *Salmonella enterica* ATCC-10708 | 15.62 | 15.62 |

**Table 4. Comparative study of biofilm formation by prodigiosin treated and untreated bacteria after 48 hours of incubation.**

| Bacteria | Absorbance at 450 nm | | p-value | % Biofilm inhibition |
|---|---|---|---|---|
| | Control | Experimental | | |
| *Listeria monocytogens* ATCC-3193 | 0.77±0.07 | 0.38±0.06 | 0.002 | ≥50% |
| *Staphylococcus aureus* | 0.70±0.03 | 0.52±0.09 | 0.03 | ≤50% |
| *Pseudomonas aeruginosa* ATCC -9027 | 0.89±0.06 | 0.45±0.06 | 0.001 | 50% |
| *Salmonella enterica* ATCC-10708 | 0.50±0.13 | 0.49±0.11 | 0.91 | Not-significant |
| *E. coli* | 0.47±0.06 | 0.27±0.05 | 0.001 | ≤50% |

These data are the average of three independent studies presented with standard deviation (n = 3). P value ≤ 0.05 was considered as significant difference that was calculated by one-way ANOVA.

Under ambient conditions, prodigiosin productivity of isolate BRL41 was 564.74 units /cell and this accumulation rate may vary significantly from strain to strain. When cultured on optimum production conditions, pigment accumulation was 36.9 units/cell for a soil isolate of *Serratia marcescens* [37] and 891.61 units/cell for *Serratia marcescens* Bizio ATCC 274TM (Sma 274) [15]. This strain to strain variations in the titer of pigment are due to the fact that these isolates may synthesize different derivatives of prodigiosin depending upon the culture conditions and presence of specific gene clusters in their genome [38]. Being an intracellular metabolite, prodigiosin extraction requires the lysis of bacterial cells and polar solvents such as ethanol, methanol, DMSO are the best choice here. The characteristic feature of HCl to take part in cell wall lysis facilitated better pigment extraction by 0.5% acidified methanol than

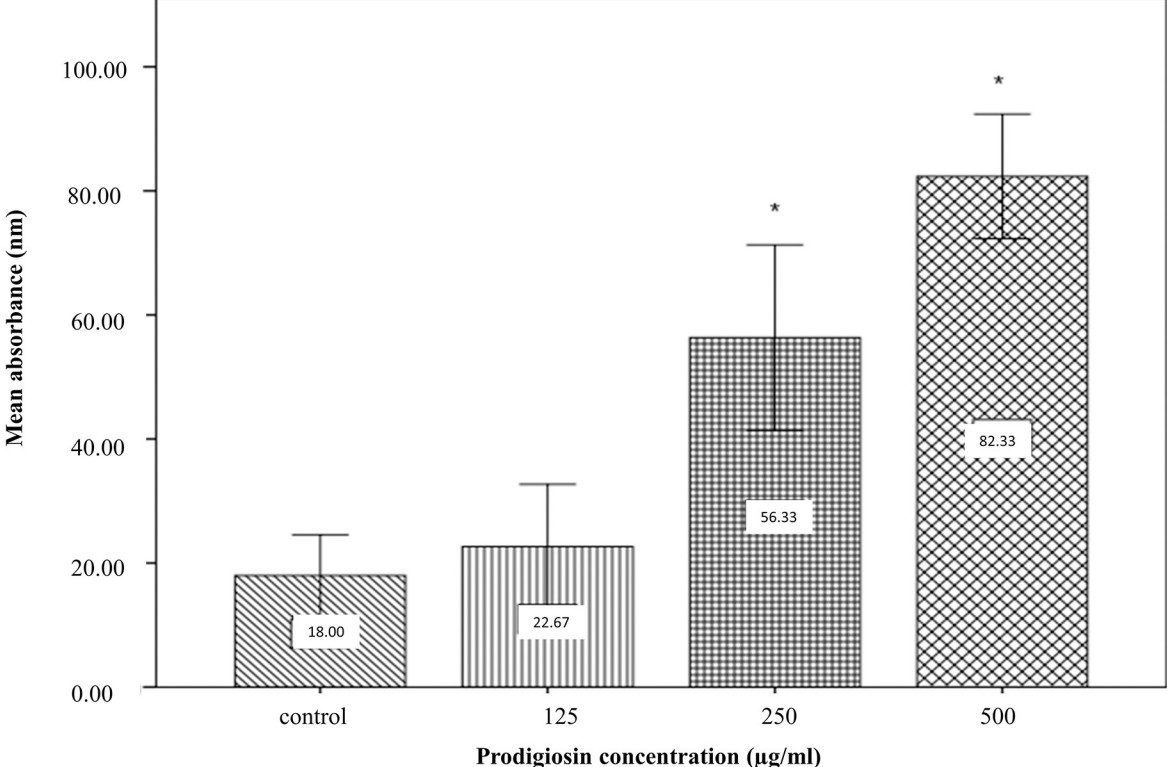

**Fig 8. Effect of prodigiosin in increasing GSH level of blood cell.** Data are presented as mean±standard deviation. *p value < 0.05 represents significant difference with control.

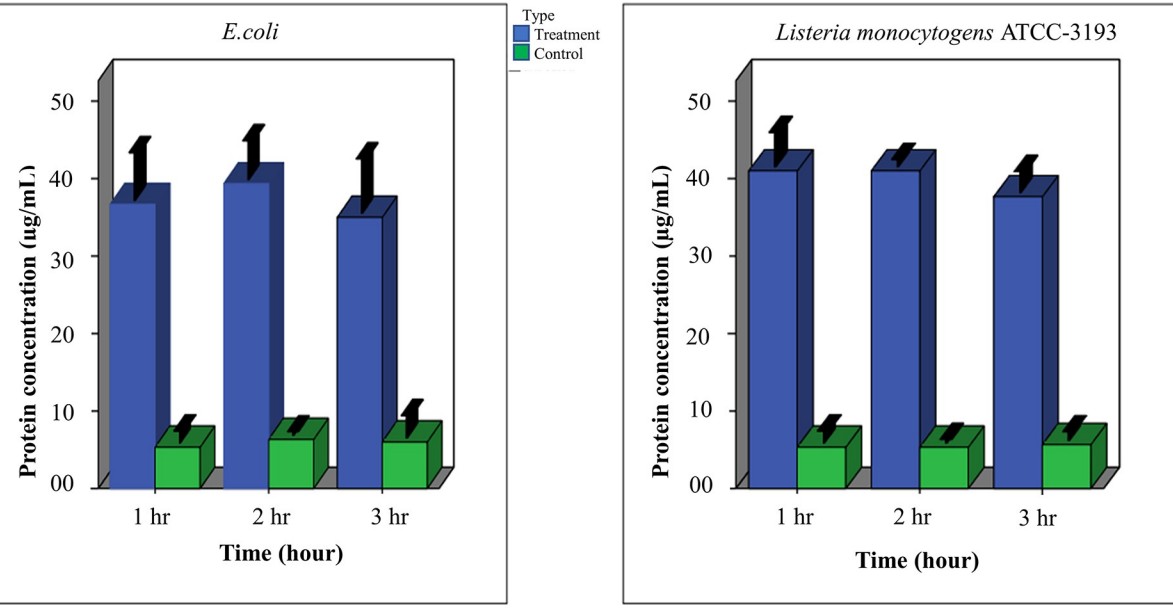

**Fig 9. Protein release from prodigiosin (500 μg/mL) treated and untreated A) Gram-positive and B) Gram-negative cell.**

methanol along. It is reported that compared to other linear molecules, the cyclic structure and pyrrole ring of prodigiosin add additional advantages in exhibition of its distinct antimicrobial activities against Gram positive and Gram-negative bacteria [39, 40]. Here, the study found prodigiosin as a promising antimicrobial agent that at high concentrations (500 μg/mL and 250 μg/mL) significantly inhibited the growth of several bacteria, an observation noticed in earlier studies [41, 42]. The failure of concentration 125 μg/mL to arrest the growth of Gram-negative *E. coli*, *Pseudomonas aeruginosa* ATCC- 9027, *Salmonella enterica* ATCC-10708 apprised prodigiosin's selectivity towards Gram-positive at low concentration. This type of resistance could be attributed by the production of biosurfactant that served as a strong emulsifier of hydrophobic compound like prodigiosin at low concentration [43]. Observation was also made on the fact that whether the studied bacteria responded differently on same concentration or not and statistical analysis displayed a significant difference in the result. This is possibly due to the variation in the degree of inhibitory effect of prodigiosin on the growth of various bacterial species. Except in *Pseudomonas aeruginosa* ATCC-9027, the MIC values for all isolates were identical (15.62 μg/mL) while the bactericidal effects were seen when 15.62 μg/mL or above concentrations were applied. Prodigiosin from *Vibrio ruber* DSM14379 towards *E. coli* had an MIC of 103.4 ± 6.3 μg/mL [44] and from *Serratia* sp. PDGS 120915 had an MIC value of 32 μg/mL against methicillin resistance *Staphylococcus aureus* [45]. These comparatively lower values of MIC and MBC suggest the competitive advantages of BRL41 prodigiosin over others to be used in the formulation of antimicrobial drug. Among 4 distinct steps in biofilm formation, microbial adhesion is considered as the first and most crucial steps. These adhesion leads to biofilm formation that imposes serious challenge in healthcare-associated infections, particularly those involving medical devices implementation [46]. Therefore, the current researches in the medical field are oriented towards bacterial adhesion prevention rather than biofilm eradication [47]. To assess the competency of prodigiosin in inhibiting microbial adhesion to growing surface, biofilm formation in prodigiosin treated and untreated bacteria was measured. Significant reduction of biofilm formation indicated reduced adherence of planktonic cell to the surface of the wall and concomitant depletion of quorum sensing

that ultimately limited biofilm formation. Exception was found in *Salmonella enterica* ATCC-10708 displaying no change in biofilm formation upon treatment with prodigiosin. Our treatment showed that *Salmonella enterica* ATCC-10708 showed significant zone of inhibition only at high concentration (500 μg/mL) with a high MIC value (15.62 μg/mL). This requirement of high concentration could be due to biofilm formation which suggests that biofilm in this bacterium can serves as an effective protective barrier against prodigiosin. Besides, the compact association of cells to the extracellular matrix inhibits the penetration of the metabolite inside the cell [48], one possible reason to impede a laser extend biofilm formation of *Staphylococcus aureus* and *Salmonella enterica* ATCC-10708. A significant reduction of biofilm formation of *Bacillus subtilis* and *Bacillus cereus* were observed when prodigiosin was applied [49]. Another aspect of this study was to appraise the value of prodigiosin as antioxidant by measuring cellular GSH level. Glutathione is a special antioxidant in cell that significantly reduces in certain diseases where oxidative stress plays the pathogenic role [50]. From the growing interest in finding a solution for increasing GSH level to treat or prevent certain diseases, we for the first time observed prodigiosin's role in upregulating GHS synthesis as an antioxidant stimulating agent though further studies are required to verify the exact role and mechanisms through which prodigiosin act as antioxidant stimulator. This positive finding of this research inspires to apply it as a therapy to diminish the onset of ageing and related diseases. For prodigiosin, a hydrophobic stressor is responsible to disrupt cellular plasma membrane via a chaotropicity -mediated- mode-of-action. Therefore, when introduced into cell, prodigiosin immediately inserts into the cell membrane and disrupt its integrity, leading to leakage of protein and ATP [51]. Herein, after prodigiosin application, measurement of cell free protein in *Listeria monocytogenes* ATCC-10708 and *E. coli* culture broth also support this finding. In both Gram-positive and Gram-negative culture, protein concentrations followed an upward trend with time and then decreased slightly. This could be due to the production of surfactant by competent cells that formed strong complexes with the Bradford reagent, making it unavailable for protein binding and interfering with the result [52]. A further investigation is required to observe the interaction between pigment and membrane to support this evidence.

Overall, the findings of our research shed a light on the applications of prodigiosin from BRL41 as an important pharmaceutical compound and further studies for better understanding of molecular mechanisms underlying the inhibitory effect of prodigiosin against pathogenic bacteria.

## Conclusion

In a nutshell, the study demonstrates the presence of metabolite biosynthesis gene clusters, production, optimization and characterization of prodigiosin by a newly isolated local *Serratia* sp. along with evaluation of its some biological and pharmaceutical characteristics. Among others, the dominated gene cluster in BRL41 is "pig cluster" whose predicted product was prodigiosin and showed similarity with non-ribosomal peptide synthases (NRPS). Prodigiosin has proved antagonist against Gram-positive and Gram-negative bacteria where one of the possible mechanisms can be the disruption of membrane integrity. It has found to function as a significant biofilm inhibitor of several pathogenic bacteria that ease cellular adhesion to surface. Increase of intracellular GSH level is a proof towards prodigiosin's contributions as an antioxidant stimulating agent. Moreover, absence of antibiotic resistance and virulence genes in chromosome of BRL41 verified its eligibility to be used for commercial production of prodigiosin. These overall findings suggest that prodigiosin from BRL41can be used for the development of drug to treat numerous diseases and infections. Besides, identification of core and regulatory genes in BGC can serve as a fundament for future improvement and modification of prodigiosin production by BRL41.

## Author Contributions

**Conceptualization:** Md. Murshed Hasan Sarkar.

**Data curation:** Farhana Boby.

**Formal analysis:** Farhana Boby, Anik Kumar Saha, Md Jahidul Islam, Mahci Al Bashera, Shyama Prosad Moulick, Farhana Jahan, Sanjana Fatema Chowdhury, Showti Raheel Naser.

**Funding acquisition:** Barun Kanti Saha, Md. Salim Khan.

**Methodology:** Farhana Boby, Farhana Jahan, Md. Murshed Hasan Sarkar.

**Project administration:** Md. Murshed Hasan Sarkar.

**Resources:** Md. Asad Uz Zaman.

**Software:** Anik Kumar Saha, Sanjana Fatema Chowdhury, Showti Raheel Naser.

**Supervision:** Barun Kanti Saha, Md. Salim Khan.

**Validation:** Shyama Prosad Moulick.

**Visualization:** Subarna Sandhani Dey, Md. Murshed Hasan Sarkar.

**Writing – original draft:** Farhana Boby.

**Writing – review & editing:** Md. Nurul Huda Bhuiyan.

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
