## [Decision Letter · Decision Letter 0]

20 Mar 2023

PONE-D-23-01982In silico exploration of Serratia sp. BRL41 genome for detecting prodigiosin biosynthetic gene cluster (BGC) and in vitro antimicrobial activity assessment of secreted prodigiosinPLOS ONE

Dear Dr. Boby,

Thank you for submitting your manuscript to PLOS ONE. After careful consideration, we feel that it has merit but does not fully meet PLOS ONE’s publication criteria as it currently stands. Therefore, we invite you to submit a revised version of the manuscript that addresses the points raised during the review process.

Authors need to add further investigations if they want to provide advanced findings, in addition to reorganizing and considering rewriting to increase the originality of the work, constructing more elaborate figures, and making statistics for comparing significantly different results. The manuscript must be completely revised. Please submit your revised manuscript by May 04 2023 11:59PM. If you will need more time than this to complete your revisions, please reply to this message or contact the journal office at plosone@plos.org. Please include the following items when submitting your revised manuscript:A rebuttal letter that responds to each point raised by the academic editor and reviewer(s). You should upload this letter as a separate file labeled 'Response to Reviewers'.A marked-up copy of your manuscript that highlights changes made to the original version. You should upload this as a separate file labeled 'Revised Manuscript with Track Changes'.An unmarked version of your revised paper without tracked changes. You should upload this as a separate file labeled 'Manuscript'.

We look forward to receiving your revised manuscript.

Kind regards,

Marcos Pileggi, Ph.D

Academic Editor

PLOS ONE

“All financial support for this work was provided by Bangladesh Council of Scientific and Industrial Research (BCSIR), Dhaka, Bangladesh.”

Reviewers' comments:

Reviewer's Responses to Questions

**Comments to the Author**

1. Is the manuscript technically sound, and do the data support the conclusions?

Reviewer #1: Partly

2. Has the statistical analysis been performed appropriately and rigorously? 

Reviewer #1: No

3. Have the authors made all data underlying the findings in their manuscript fully available?

Reviewer #1: Yes

4. Is the manuscript presented in an intelligible fashion and written in standard English?

Reviewer #1: No

5. Review Comments to the Author

Reviewer #1: The submitted paper

“In silico exploration of Serratia sp. BRL41 genome for detecting prodigiosin biosynthetic gene cluster (BGC) and in vitro antimicrobial activity assessment of secreted prodigiosin

Describes in vitro antimicrobial activity of secreted prodigiosin FROM Serratia sp. and the synthesis of prodigiosin from Serratia is a well known process. Also, in silico exploration of Serratia sp. BRL41 biosynthetic gene cluster (BGC).

This work lacks originality as most findings were described elsewhere (Production of prodigiosin by S marcescens. it's inhibitory and antibiofilm effects..etc). Other investigations should be added if the authors want to give advanced findings. (for example, discovery of new biological or pharmacological effects..). In addition, the authors should rearrange, and consider rewriting in order to rise originality of the work, consider Figure relevance for publication and do statistics for comparison of significant different results.

I suggest this before resubmitting a revised version.

Other remarks

Line19 : E.coli : Please to write complete name of the species

What is type strains of “Staphylococcus aureus”.

Line 22-24: finding not clear, please to support briefly by arguments

Line 79-84: Please to formulate clearly ideas specifically relationships and outcomes regarding BGC and prodigiosin. Please to give clear originality of the work. Please to focus on outcomes and significance of your results rather than describing methods

In the absence of originality, the MS is not suitable to be published in the PLOS journal

Line 104: give manufacturer name

Line107: give reference for “gene assembly by Shovill pipeline ».

Line 146-147: please to rewrite

Please to add references for techniques in M&M.

Line 286: Write “s” in “staphylococcus » in capital letter

Line 227: please write “sakeunsis” in italic

Figure 1: Gram reaction of isolate BRL41 There is no need for this figure, as simple description in the text is sufficient

Figure 2: There is no need for this display, or to move to Appendix.

Figure 5: There is no need for this figure (part showing color change), as simple description in the text is sufficient

Conclusion:

Please to avoid redundancies of the last discussion paragraph. Overall avoid a conclusion that resembles a summary.

6. PLOS authors have the option to publish the peer review history of their article (what does this mean?). If published, this will include your full peer review and any attached files.

Reviewer #1: No

---

## [Author Response · Author response to Decision Letter 0]

16 May 2023

Response/Action: Thanks to the editor for this nice suggestion. Following his comment, we have corrected the format of our manuscript according to PLOS ONE’s requirements. We have also changed the file name here.

Before revision:

File name: Manuscript for Plos one

After revision:

File name: Manuscript. Please go through the file “Revised Manuscript with Track Changes.docx”

 Response/Action: Thank you for this comment

 Before revision: 

“All financial support for this work was provided by Bangladesh Council of Scientific and Industrial Research (BCSIR), Dhaka, Bangladesh.”

After revision:

All financial support for this work was provided by Bangladesh Council of Scientific and Industrial Research (BCSIR), Dhaka, Bangladesh. As it is a Government organization and I am an employee here, BCSIR has provided me with fund but it has no role in study design, data collection and analysis, decision to publish or preparation of manuscript. This statement has also included in “cover letter”.

Reviewer’s comments:

 Is the manuscript technically sound, and do the data support the conclusions?

Reviewer: Partly

Response/Action: Thank you for your comment. Based on reviewers’ comment, we have modified the conclusion based on the data so that it can support the research. All works have done in triplicate set with appropriate control where necessary.

Before revision:

Conclusion-This study concentrated on the isolation, characterization, and evaluation of antimicrobial activity of prodigiosin synthesized by newly isolated bacterium from the soil of BCSIR Rajshahi Laboratories, Bangladesh. This investigation identified the isolated bacteria as Serratia sp. BRL41 harboring the prodigiosin biosynthetic gene cluster. Prodigiosin synthesis by isolate BRL41 attained its maxima at 30˚C temperature, pH 7.5 after 96 hours of incubation. It appeared as a potential biological agent inhibiting bacterial growth and biofilm formation by lysing bacterial cell walls. Thus, this study may aid the development of an antimicrobial drugs from prodigiosin to combat bacterial infection. Additionally, it can encourage researchers to work with the prodigiosin in detail to determine its mode of action.

After revision:

Conclusion- In a nutshell, our study demonstrates the production, optimization and characterization of prodigiosin by a newly isolated local Serratia sp. along with evaluation of its some biological and pharmaceutical characteristics. Prodigiosin has found antagonist against Gram-positive and Gram-negative bacteria where the possible mechanism can be the cell wall disruption or production of reactive oxygen species. Prodigiosin also reduces biofilm formation possibly by modulating virulence gene expression, aggregation, hydrophobicity, acid production, acid tolerance. Increase of intracellular glutathione is a proof towards prodigiosin’s antioxidant activity. These findings suggest that prodigiosin can be used for the development of antimicrobial drug that can combat bacterial infection. In addition, identification of core and regulatory genes in BGC can work as a fundament for future improvement and modification of prodigiosin production by BRL41.Besides, further studies are required for better understanding of molecular mechanisms underlying the inhibitory effect of prodigiosin against pathogenic bacteria.

 Has the statistical analysis been performed appropriately and rigorously?

Reviewer: No

Response/ Action: Thank you for reviewer’s great comment. Following reviewers’ comment, we have analyzed all data using IBM SPSSS statistics 22 software. All data are expressed as mean± standard deviation where p value was calculated at 95% significant level. Please go through the file “Revised Manuscript with Track Changes.docx”

Before revision:

Statistical analysis: No information available

After revision:

Statistical analysis was performed using IBM SPSSS statistics 22 software. All data were expressed as mean ± standard deviation. One-way ANOVA followed by Tukey test was conducted for comparing means of zone of inhibition. p value < 0.05 were considered to be statistically significant at 95% confidence level.

 Have the authors made all data underlying the findings in their manuscript fully available?

Reviewer: Yes

Response/Action: Thank you for reviewer’s great comment.

 Is the manuscript presented in an intelligible fashion and written in standard English?

Reviewer: No

Response/Action: With some data rearrangement and change of language, we have tried our best to present the manuscript in an intelligible fashion with simple English that can become easily understandable to all. Please have a look at file “Revised Manuscript with Track Changes.docx”

 This work lacks originality as most findings were described elsewhere (Production of prodigiosin by S marcescens. it's inhibitory and antibiofilm effects. etc). Other investigations should be added if the authors want to give advanced findings. (for example, discovery of new biological or pharmacological effects). In addition, the authors should rearrange, and consider rewriting in order to rise originality of the work, consider Figure relevance for publication and do statistics for comparison of significant different results. 

Response/Action: Thank you for reviewer’s comment and suggestions. Many works have been conducted on prodigiosin in different parts of the world but in Bangladesh this work has done for the first time with a new isolate obtained from Barind soil (a special type of native soil in Rajshahi region of Bangladesh).Following the suggestion of reviewer we have analyzed the zone of inhibition data in a different way to show whether change of prodigiosin concentrations affect antimicrobial activity or not. In addition, we have explored a new pharmaceutical activity of prodigiosin i.e cellular reduced glutathione level (GSH) measurement which is, to the best of our knowledge, the first work with prodigiosin. Besides, we have changed figure 9 (Protein release assay) and presented it in a more acceptable fashion.

Before revision:

Bacteria Diameter of Inhibition mm

 Positive Control Negative Control Prodigiosin

 Chloramphenicol

(30µg/mL) DMSO 500 µg/mL 250 

µg /mL 125

 µg /mL

E coli 26±1.00 0 21±0.60 17±0.69 No zone

Staphylococcus aureus 28±0.58

 0 19±0.56 14±0.76 9±0.00

Listeria monocytogens ATCC-3193 28.67±0.58 0 19±0.10 13±0.34 10±1.15

Pseudomonas aeruginosa ATCC- 9027 26±0.58 0 25±0.25 17±0.20 No zone

Salmonella enterica ATCC-10708 25±0.58 0 20±0.09 8±0.75 No zone

After revision:

 Concentration Tested bacteria

 (µg/ml) E. coli Staphylococcus aureus Listeria monocytogens ATCC-3193 Pseudomonas aeruginosa ATCC- 9027 Salmonella enterica ATCC-10708

Prodigiosin 500 25.61 ± 0.15a 18.66 ± 0.0.33b 18.56 ± 0.22b 24.73 ± 0.15b 19.60 ± 0.05b

 250 16.89 ± 0.40b 14.00 ± 0.45c 13.00 ± 0.20c 16.61 ± 0.11c 7.77 ± 0.39b

 125 No zone 9.00 ± 0.00d 9.00 ± 0.00d No zone No zone

 +ve (control) 26.00 ± 0.58a 27.67 ± 0.33a 27.67 ± 0.33a 26.33 ± 0.33a 25.00 ± 0.00a

Before revision:

No data about GSH content measurement of blood cell after exposing on prodigiosin

After revision:

Prodigiosin increased cellular GSH level 

Over the range of 125-500 µg/ml, prodigiosin significantly increased (p< 0.05) reduced glutathione level (GSH) in cell at 250 and 500 µg/ml concentration “Fig 8”. Prodigiosin at these concentrations stimulated glutathione synthesis that proved its efficacy as an antioxidation stimulating agent

Fig 8: Effect of prodigiosin on increasing GSH level of blood cell.Data are presented as mean±standard deviation. *p value < 0.05 represents significant difference with control

Before revision:

Fig 9: Protein release from prodigiosin (500 µg/mL) treated and untreated A) Gram-positive and B) Gram-negative cell

After revision:

Fig 9: Protein release from prodigiosin (500 µg/mL) treated and untreated A) Gram-positive and B) Gram-negative cell

Other remarks:

Line19: E. coli: Please to write complete name of the species

What is type strains of “Staphylococcus aureus”.

Action: Amended

Before revision:

E.coli, Staphylococcus aureus,

After revision:

Escherichia coli, Staphylococcus aureus (environmental isolate)

Line 22-24: finding not clear, please to support briefly by arguments

Action: Thank you for this comment. For better understanding and to support our arguments we have tried to explain the work more clearly than before.

Before revision:

Furthermore, the increase of protein concentration in prodigiosin treated bacterial suspension was an evidence of cell lysis resulting in protein release from cells.

After revision:

Cell wall as one of the sites of action of prodigiosin was manifested by the increase of protein concentration on prodigiosin treated cell suspension that was resulted from protein leakage by damaged cell wall over time.

Line 79-84: Please to formulate clearly ideas specifically relationships and outcomes regarding BGC and prodigiosin. Please to give clear originality of the work. Please to focus on outcomes and significance of your results rather than describing methods. In the absence of originality, the MS is not suitable to be published in the PLOS journal

Action: We have re-written this section with adding more relevant informations.

Before revision: 

So, the study observed the behavior of prodigiosin towards both Gram-positive and Gram-negative bacterial pathogens. Firstly, Serratia sp. BRL41 with prodigiosin-producing capability was isolated from soil in BCSIR, Rajshahi, Bangladesh. Prodigiosin was produced and extracted using solvent extraction method followed by its identification using Thin-layered Chromatography (TLC), UV-spectroscopy, Fourier Transform Infrared Spectroscopy (FTIR), and pH dependent color change. This study also focused on finding the optimum conditions for bacterial prodigiosin biosynthesis and extraction. Another aspect of this study was to utilize the whole genome sequence data of the isolate to identify the Biosynthetic Gene Clusters (BGC) associated with secondary metabolite production. Antimicrobial activity of prodigiosin was tested by agar well diffusion method and minimum inhibitory concentration (MIC) and minimum bactericidal concentration (MBC) assay. Biofilm formation inhibition and protein release capability of prodigiosin were also explored here. The outcomes give an evidence of prodigiosin’s bactericidal efficacy that facilitates its survival at competitive environment as well as explore a potential bacterial metabolite that can be used as alternative to antibiotic.

After revision:

So, the study addressed the bacteriostatic/bactericidal behavior of prodigiosin towards both Gram-positive and Gram-negative bacterial pathogens. Serratia sp. BRL41 is the first reported prodigiosin-producing bacteria in Bangladesh isolated from barhind soil. Prior to checking antimicrobial activity,prodigiosin was identified and characterized. Another aspect of this study was to utilize the whole genome sequence data of the isolate to identify the Biosynthetic Gene Clusters (BGC) associated with secondary metabolite biosynthesis that facilitated the understanding of prodigiosin biosynthesis and regulation mechanism with the identification of core biosynthetic gene in BRL41. For the first time, we have investigated membrane integrity of Gram-positive and Gram-negative bacterial cell upon treated with prodigiosin from Serratia marcescens as well as role of prodigiosin as an anti-oxidant by measuring intracellular reduced glutathione level (GSH) of human blood cell. In a nutshell, in search of new antimicrobial molecules, prodigiosin’s biological and pharmaceutical efficacies provides evidence of its potency as an alternative to antibiotics.

Line 104: Give manufacturer name

Action: Manufacturer’s name of Wizard® Genomic DNA Purification Kit has added.

Before revision:

DNA from the BRL41 isolate was extracted using Wizard® Genomic DNA Purification Kit and whole genome sequencing was done by Illumina MiniSeq platform.

After revision:

DNA from the BRL41 isolate was extracted using Wizard® Genomic DNA Purification Kit (Promega, catalog no. PR-A1120) and whole genome sequencing was done by Illumina MiniSeq platform.

Line107: give reference for “gene assembly by Shovill pipeline ».

Action: Reference of Shovill pipeline has added here.

Before revision:

Library preparation was conducted as per the manufacturer’s protocol. After obtaining the raw data, the assembly of the genome was performed using the Shovill pipeline. This genome assembler utilized SPAdes as its core and is specific for bacterial isolated genomes from Illumina paired end reads.

After revision:

After obtaining the raw data, the assembly of the genome was performed using the Shovill pipeline (GitHub - Tseemann/Shovill: Assemble Bacterial Isolate Genomes from Illumina Paired-End Reads, n.d.)This genome assembler utilized SPAdes as its core and is specific for bacterial isolated genomes from Illumina paired end reads.

Line 146-147: please to rewrite

Action: The sentence has corrected. In addition, we have re-written this section.

Before revision:

Optimum temperature for pigment biosynthesis was determined by inoculating nutrient broth with bacteria was incubating at 25°C, 28°C, 30°C, 37°C temperatures. After 96 hours of incubation, the color intensity was measured spectrophotometrically at 537 nm and the result was recorded. One of the crucial factors for pigment production, the pH, was fixed by allowing BRL41 to grow at different pH nutrient broth ranged from 6.00 to 8.50. Pigment production at each pH medium was quantified to determine the optimum pH for maximum prodigiosin production.

After revision:

To find out optimum conditions for maximum prodigiosin production by Smi41, some important parameters like incubation time (24, 48, 72, 96 hour), temperature (25, 28, 30, 37˚ C), pH (5.00 to 8.5) were checked using nutrient broth as basal media. After each treatment color intensity was measured spectrophotometrically at 537 nm where the highest absorbance signified the highest pigment production on that particular condition. 

Please to add references for techniques in M&M.

Action: Thank you for this great suggestion. We have added references for all the techniques used to conduct this work.

Before revision:

Phylogenetic and Biosynthetic Gene Cluster (BGC) analysis

DNA from the BRL41 isolate was extracted using Wizard® Genomic DNA Purification Kit and whole genome sequencing was done by Illumina MiniSeq platform. Library preparation was conducted as per the manufacturer’s protocol. After obtaining the raw data, the assembly of the genome was performed using the Shovill pipeline. This genome assembler utilized SPAdes as its core and is specific for bacterial isolated genomes from Illumina paired end reads. Phylogenetic analysis was performed using Type (Strain) Genome Server (TYGS) using maximum-likelihood method (21). The phylogenetic tree was visualized using iTOL(22). Biosynthetic gene clusters responsible for the secondary metabolite prodigiosin production were identified using antiSMASH (23). This identified bacterial isolate was used to conduct further studies.

Determination of Minimum Inhibitory Concentration (MIC) and Minimum Bactericidal Concentration (MBC)

MIC of prodigiosin was assessed against these five species of bacteria by dilution method in test tubes. Firstly, the optical density of 24 hours culture of each bacterium was adjusted to 0.4-0.6. Then a solution of prodigiosin with a final concentration of 500µg/mL was prepared using 10% dimethyl sulfoxide (DMSO). 1ml of prodigiosin was added in the 1st tube and then was serially diluted to obtain concentrations ranging from 500µg/mL to 0.24µg/mL. Tube containing culture broth and inoculum represented a positive control whereas a tube with culture broth, inoculum, and antibiotic chloramphenicol was used as a negative control. These were incubated at 30°C overnight. Then the growth pattern of bacteria was observed to determine the minimum inhibitory concentration based on the absence of visible growth of each bacterium. MBC was determined by spreading 10 µL of culture from each tube on nutrient agar plate to find out the concentration that caused the complete death of bacterial cells. Concentration for which no bacterial growth was observed on the plate was considered as MBC.

 Anti-biofilm activity of prodigiosin

The ability of prodigiosin to inhibit biofilm formation of pathogenic bacteria was evaluated by crystal violet (CV) assay in 96 well microtiter plates. A serial dilution of prodigiosin was made using 10% DMSO in a microtiter plate and bacterial culture was then added to every well to make a final volume of 100 µL. For every bacterium, nutrient broth represented blank, whereas a mixture of nutrient broth, DMSO, and 10 µL of bacteria was used as control. After 48 hours of incubation, each well was washed twice with 1×PBS solution followed by the addition of 200µL of 99% methanol. It was kept for 20 minutes and then the methanol was discarded. After drying the plate for 15 minutes, 200 µL of 0.1% crystal violet was added to each well and kept at room temperature for 5 minutes so that it could stain the bacterial cell. Crystal violet was drained out and the plate was dried completely. Finally, to solubilize the crystal violet attached to the bacterial cell, 95% ethanol was added and mixed. Then the optical density was measured at 450nm using a microtiter plate reader. Biofilm inhibition % was calculated using the following formula (25)

(〖(OD〗_control-OD_treated))/OD_control ×100

After revision:

Phylogenetic and Biosynthetic Gene Cluster (BGC) analysis

DNA from the BRL41 isolate was extracted using Wizard® Genomic DNA Purification Kit (Promega, catalog no. PR-A1120) and whole genome sequencing was done by Illumina MiniSeq platform. Library preparation was conducted as per the manufacturer’s protocol. After obtaining the raw data, the assembly of the genome was performed using the Shovill pipeline (21).This genome assembler utilized SPAdes as its core and is specific for bacterial isolated genomes from Illumina paired end reads. Phylogenetic analysis was performed by Type (Strain) Genome Server (TYGS) using maximum-likelihood method (22). The phylogenetic tree was visualized using iTOL(23). Biosynthetic gene clusters responsible for the secondary metabolite prodigiosin production were identified using antiSMASH (24). This identified bacterial isolate was used to conduct further studies.

Determination of Minimum Inhibitory Concentration (MIC) and Minimum Bactericidal Concentration (MBC)

MIC of prodigiosin was assessed against these five species of bacteria by macro dilution method in test tubes (26). Firstly, the optical density of 24 hours culture of each bacterium was adjusted to 0.4-0.6. Then a solution of prodigiosin with a final concentration of 500µg/mL was prepared using 10% dimethyl sulfoxide (DMSO). 1ml of prodigiosin was added in the 1st tube and then was serially diluted to obtain concentrations ranging from 500µg/mL to 0.24µg/mL. Tube containing culture broth and inoculum represented a positive control whereas a tube with culture broth, inoculum, and antibiotic chloramphenicol was used as a negative control. These were incubated at 30°C overnight. Then the growth pattern of bacteria was observed to determine the minimum inhibitory concentration based on the absence of visible growth of each bacterium. MBC was determined by spreading 10 µL of culture from each tube on nutrient agar plate to find out the concentration that caused the complete death of bacterial cells. Concentration for which no bacterial growth was observed on the plate was considered as MBC.

Anti-biofilm activity of prodigiosin

The ability of prodigiosin to inhibit biofilm formation of pathogenic bacteria was evaluated by crystal violet (CV) assay in 96 well microtiter plates (27). A serial dilution of prodigiosin was made using 10% DMSO in a microtiter plate and bacterial culture was then added to every well to make a final volume of 100 µL. For every bacterium, nutrient broth represented blank, whereas a mixture of nutrient broth, DMSO, and 10 µL of bacteria was used as control. After 48 hours of incubation, each well was washed twice with 1×PBS solution followed by the addition of 200µL of 99% methanol. It was kept for 20 minutes and then the methanol was discarded. After drying the plate for 15 minutes, 200 µL of 0.1% crystal violet was added to each well and kept at room temperature for 5 minutes so that it could stain the bacterial cell. Crystal violet was drained out and the plate was dried completely. Finally, to solubilize the crystal violet attached to the bacterial cell, 95% ethanol was added and mixed. Then the optical density was measured at 450nm using a microtiter plate reader. Biofilm inhibition % was calculated using the following formula (28)

(〖(OD〗_control-OD_treated))/OD_control ×100

Antioxidant assay on by estimating reduced glutathione (GSH) content in cell

This test was performed following the methodology described by (29) with some modification. In brief, Prodigiosin of varying concentrations (125,250,500 µg/ml) were taken in three separate falcon tube and erythrocyte suspension was added and mixed by gentle inversion. Erythrocytes with PBS was served as control and all the tubes were incubated at 37˚C for 24 hours. After centrifugation of reaction mixtures, water addition facilitated cell lysis followed by addition of metaphosphoric acid that resulted in precipitation of cell lysate. Following centrifugation both control and treatment tubes were treated with 0.01 M sodium phosphate buffer (pH 7.5), 5 mM EDTA, 10 mM 5,5-dithio-bio-nitrobenzoic acid(DTNB), 2mM NADPH, 100 U/ml glutathione reductase to reach a final volume 1ml.Absorbence was taken at 412 nm against blank.

Line 286: Write “s” in “staphylococcus » in capital letter

Action: In Staphylococcus small “s”has replaced with capital “S”

Before revision:

A higher MIC (> 10µg/mL) was required to inhibit the growth of E. coli, staphylococcus aureus, and Listeria monocytogenes ATCC-3193 compared to Pseudomonas aeruginosa ATCC-9027, and Salmonella enterica ATCC-10708.

After revision:

A higher MIC (> 10µg/mL) was required to inhibit the growth of E. coli, Staphylococcus aureus, and Listeria monocytogenes ATCC-3193 compared to Pseudomonas aeruginosa ATCC-9027, and Salmonella enterica ATCC-10708.

Line 227: please write “sakeunsis” in italic

Action: The word “sakeunsis”has changed to italic

Before revision:

Sequence alignment and construction of the phylogenetic tree revealed that the isolate Serratia sp. BRL41 is very closely related to strain Serratia marcescens subsp., sakeunsis KCTC 42172, and Serratia marcescens ATCC 13880

After revision:

Sequence alignment and construction of the phylogenetic tree revealed that the isolate Serratia sp. BRL41 is very closely related to strain Serratia marcescens subsp., sakeunsis KCTC 42172, and Serratia marcescens ATCC 13880

Figure 1: Gram reaction of isolate BRL41 There is no need for this figure, as simple description in the text is sufficient

Action: Based on the comment of reviewer’s, Gram reaction figure of isolate BRL41 has delated from revised manuscript.

Figure 2: There is no need for this display, or to move to Appendix.

Action: Thank you for your suggestion. Figure 2 of original manuscript has been delated in revised one

Figure 5: There is no need for this figure (part showing color change), as simple description in the text is sufficient

Action: Thanks for reviewer’s comment. Based on it, the figure has corrected by deleting color changing part and kept only the TLC section.

Before action: 

After action:

---

## [Decision Letter · Decision Letter 1]

22 Jun 2023

PONE-D-23-01982R1In silico exploration of Serratia sp. BRL41 genome for detecting prodigiosin biosynthetic gene cluster (BGC) and in vitro antimicrobial activity assessment of secreted prodigiosinPLOS ONE

Dear Dr. Boby,

Thank you for submitting your manuscript to PLOS ONE. After careful consideration, we feel that it has merit but does not fully meet PLOS ONE’s publication criteria as it currently stands. Therefore, we invite you to submit a revised version of the manuscript that addresses the points raised during the review process. Please submit your revised manuscript by Aug 06 2023 11:59PM. If you will need more time than this to complete your revisions, please reply to this message or contact the journal office at plosone@plos.org. Please include the following items when submitting your revised manuscript:A rebuttal letter that responds to each point raised by the academic editor and reviewer(s). You should upload this letter as a separate file labeled 'Response to Reviewers'.A marked-up copy of your manuscript that highlights changes made to the original version. You should upload this as a separate file labeled 'Revised Manuscript with Track Changes'.An unmarked version of your revised paper without tracked changes. You should upload this as a separate file labeled 'Manuscript'.

We look forward to receiving your revised manuscript.

Kind regards,

Marcos Pileggi, Ph.D

Academic Editor

PLOS ONE

**Additional Editor Comments:**

Authors need to address all reviewers' comments as the revised version of the manuscript still has some important issues that will restrict its publication in Plos One.

Reviewers' comments:

Reviewer's Responses to Questions

**Comments to the Author**

1. If the authors have adequately addressed your comments raised in a previous round of review and you feel that this manuscript is now acceptable for publication, you may indicate that here to bypass the “Comments to the Author” section, enter your conflict of interest statement in the “Confidential to Editor” section, and submit your "Accept" recommendation.

Reviewer #2: All comments have been addressed

Reviewer #3: (No Response)

Reviewer #4: (No Response)

2. Is the manuscript technically sound, and do the data support the conclusions?

Reviewer #2: Partly

Reviewer #3: Partly

Reviewer #4: Yes

3. Has the statistical analysis been performed appropriately and rigorously? 

Reviewer #2: Yes

Reviewer #3: I Don't Know

Reviewer #4: Yes

4. Have the authors made all data underlying the findings in their manuscript fully available?

Reviewer #2: No

Reviewer #3: Yes

Reviewer #4: Yes

5. Is the manuscript presented in an intelligible fashion and written in standard English?

Reviewer #2: No

Reviewer #3: Yes

Reviewer #4: No

6. Review Comments to the Author

Reviewer #2: The authors addressed all the comments from the previous review, however the revised version of the manuscript still retains some major issues that will restrict its publication in Plos One. The manuscript lacks novelity and more than 80% of the results are already established by previous studies. Authors tried to describe the elavated GSH activity as a novel property, but the discussion section is incomplete regarding its consequences and significance with the present study.

Authors also claimed in the response to reviewer section that prodigiosin affects the membrane integrity in gram positive and gram negative bacteria, which is the first report; but effect of prodigiosin on cell membrane integrity has been described previously through fluorescence microscopy and ion leakage assay.

Authors mentioned WGS of the bacterial isolate, but did not provided the access to those results. As per the journal policy, any molecular data generated should be deposited to public database and accession numbers to be mentioned in the manuscript.

Additionally, authors tried improved the english language use and sentence writing skills in the revised version, however there are still some major issues with grammars and sentence construction, which need to be checked properly.

Overall, from my point of view, authors failed to meet the publication standard of this journal. I suggest them to address the aforementioned issues, and the results and discussion sections should focus on identifying the mechanism of action of prodigiosins in the target pathogens.

Reviewer #3: The manuscript by Boby et al. describes the selection of a prodigiosin producing strain of Serratia marcescens. Authors propose the use of this metabolite as an antibacterial agent to overcome the problem of antimicrobial resistance. Overall, the topic is of great interest however, the manuscript still need substantial revision before being acceptable for publication.

General comments:

-The selection of the strain still need some clarifications. What they did was based on the screening of pigmented strains or search for Serratia species. The sentence in line 99-100 needs more clarification;

-References should be updated (use articles published on 2021, 2022 and 2023). there are many articles recently published focusing on prodiginines production and characterization.

-Change of color with pH, FTIR and UV are considered being sufficient to conclude about the structure of the compound? I think authors have to confirm with MS or NMR analysis.

-The mechanism of action still need further investigations to be confirmed

Minor comments:

-English should be checked carefully

-"sp" should be not itallic "sp." throughout all the manuscript

-line 126-127: "As the absorption were..........1000". THis is not convinced.

-Ethyl acetate as the only solvent for TLC analysis?

-the antibiofilm activity that has been conducted is the antiadhesive activity, what about eradication?

-The concentration of chloramphenicol used is 30µg/ml. Why the choice of chloramphenicol and how could you compare with prodigisin since the concentrations used are different.

-The equation used to quantify prodigiosin per cell should be supported with a reference

-line 57 "Researchers claim that prodigiosin is produced mostly by the non-pathogenic strain of Serratia marcescens(14)". I am not sure if the reference used supports this idea. Authors have to clarify this point.

Reviewer #4: The manuscript describes the Insilco detection of prodigiosin biosynthesis gene cluster in Serratia sp. BRL41 and, also describes the isolation and antimicrobial activity of prodigiosin against gram- positive and gram- negative bacteria. However, most of the results presented here were previously described by many researchers. Hence, novelty of this study is missing. This study also possesses some experimental drawbacks in the study design as well as discussion of the results. The authors mentioned WGS analysis of isolate BRL41 but, results and methodologies were not described in detail. Link for the WGS data is also provided. The phylogenetic analysis was performed but the accession no for BRL41 was not provided. The authors failed to describe the speciality of this isolate as the BGC of prodigiosin is similar to those previously reported. The discussion section is weakly constructed and missed many of the important citations in the relevant area. The authors should give emphasis on the mechanism of antibacterial properties and targets in the pathogens. The discussion section should emphasise on the scope for the use of prodigiosin against the target pathogens and their target mechanisms at molecular level. Additionally, the whole manuscript should be properly checked by a native English speaker for improvement of the language used. Some specific comments are given below-

1. Unit should be synchronised for e.g. The gap between the value and the unit should be maintained.

2. Line 5- should mention properly the type of soil from which the bacteria were isolated.

3. Line 39- space needed in 10µg/mL(5).

4. Line 55- space needed in marcescens(14).

5. Line 57- needed a dot in etc

6. Line 58- space needed in (15)Physicochemical.

7. Line 60- space needed so check the line.

8. Line 65- the scientific name should be in italics, check the line.

9. Line 109-110- Modify the line, reduce the use of the word ‘using’ in to one time.

10. Line 110- space needed in iTOL(22), check the line.

11. Line 124-should replace the word ‘with’ by the word ‘by’.

12. Line 149- the symbol comma is not needed after the word pH.

13. Line 162- the symbol comma in ‘cool, dry’ should be replaced by the word ‘and’.

14. Line 213- Space is needed in (26)Bovine.

15. Line 331-333- Modify the language (Therefore,……..animals).

7. PLOS authors have the option to publish the peer review history of their article (what does this mean?). If published, this will include your full peer review and any attached files.

Reviewer #2: No

Reviewer #3: No

Reviewer #4: No

---

## [Author Response · Author response to Decision Letter 1]

6 Aug 2023

Reviewers comments:

1. If the authors have adequately addressed your comments raised in a previous round of review and you feel that this manuscript is now acceptable for publication, you may indicate that here to bypass the “Comments to the Author” section, enter your conflict of interest statement in the “Confidential to Editor” section, and submit your "Accept" recommendation.

Reviewer 2: Reviewer #2: All comments have been addressed

Reviewer #3: (No Response)

Reviewer #4: (No Response)

Response/Action: Thank you for all reviewers’ great comment

2. Is the manuscript technically sound, and do the data support the conclusions?

Reviewer #2: Partly

Response/ action: Thank you for reviewer’s comment. Based on reviewers comment we have made some corrections in conclusion section to make the highest alignment between data and conclusion

Before revision: In a nutshell, our study demonstrates the production, optimization and characterization of prodigiosin by a newly isolated local Serratia sp. along with evaluation of its some biological and pharmaceutical characteristics. Prodigiosin has found antagonist against Gram-positive and Gram-negative bacteria where the possible mechanism can be the cell wall disruption or production of reactive oxygen species. Prodigiosin also reduces biofilm formation possibly by modulating virulence gene expression, aggregation, hydrophobicity, acid production, acid tolerance. Increase of intracellular glutathione is a proof towards prodigiosin’s antioxidant activity. These findings suggest that prodigiosin can be used for the development of antimicrobial drug that can combat bacterial infection. In addition, identification of core and regulatory genes in BGC can work as a fundament for future improvement and modification of prodigiosin production by BRL41.Besides, further studies are required for better understanding of molecular mechanisms underlying the inhibitory effect of prodigiosin against pathogenic bacteria.

After revision: In a nutshell, the study demonstrates the presence of metabolite biosynthesis gene clusters, production, optimization and characterization of prodigiosin by a newly isolated local Serratia sp. along with evaluation of its some biological and pharmaceutical characteristics. Among others, the dominated gene cluster in BRL41 is “pig cluster” whose predicted product is prodigiosin and shows similarity with non-ribosomal peptide synthases (NRPSs). Prodigiosin has proved antagonist against Gram-positive and Gram-negative bacteria where one of the possible mechanisms can be the disruption of membrane integrity. It has found to function as a significant biofilm inhibitor of several pathogenic bacteria that ease cellular adhesion to surface. Increase of intracellular GSH level is a proof towards prodigiosin’s contributions in as an antioxidant stimulating agent. Moreover, absence of antibiotic resistance and virulence genes in chromosome of BRL41 verified its eligibility to be used for commercial production of prodigiosin. These overall findings suggest that prodigiosin from BRL41can be used for the development of drug to treat numerous diseases and infections. Besides, identification of core and regulatory genes in BGC can serve as a fundament for future improvement and modification of prodigiosin production by BRL41. 

Reviewer #3: Partly

Response/ Action: Thank you for your comment. We have made some correction on conclusion section to make the highest alignment between data and conclusion. Please see the response of reviewer 2 in this context.

3. Has the statistical analysis been performed appropriately and rigorously?

Reviewer #2: Yes

Response/ Action: Thank you for reviewer’s great comment

Reviewer #3: I don’t know

Response/ Action: Thank you for reviewer’s comment. We have performed statistical analysis using IBM SPSSS statistics 22 software to increase the reproducibility of results. All the data are expressed as mean ± standard deviation. Also, one-way ANOVA followed by Tukey test has conducted for comparing means of zone of inhibition. p value < 0.05 were considered to be statistically significant at 95% confidence level.

Reviewer #4: Yes

Response/ Action: Thank you for reviewer’s great comment

4. Have the authors made all data underlying the findings in their manuscript fully available?

Reviewer #2: No

Response/ Action: Thank you for reviewer’s comment. We have made all the data underlying the findings available in our manuscript. Additionally, the accession number of WGS has been added in the revised version of the manuscript 

Reviewer #3: Yes

Response/Action: Thank you for reviewer’s great comment.

Reviewer #4: Yes

Response/Action: Thank you for reviewer’s great comment.

5. Is the manuscript presented in an intelligible fashion and written in standard English?

Reviewer#2: No

Response/ Action: Thank you so much for the comment reviewer has made here. We have re-written our abstract, added rationale of the study, current findings in this field with references and modified the conclusion section. A major correction has made on discussion section by giving emphasis on the scope of the study. The language and presentation style throughout the manuscript has modified so that it can be presented in a standard form suitable for publication on “PLOS ONE”. Moreover, some major data and statistical analysis in result has been adjusted for better explanation. Please see the attachment “Manuscript with track changes.docx”

 Reviewer 3: Yes

Response/ Action: Thank you for the great comment reviewer has made here.

Reviewer 4: No

Response/ Action: Thank you so much for the comment reviewer has made here. We have re-written our abstract, added rationale of the study, current findings in this field with references and modified the conclusion section. A major correction has made on discussion section by giving emphasis on the scope of the study. The language and presentation style throughout the manuscript has modified so that it can be presented in a standard form suitable for publication on “PLOS ONE”. Moreover, some major data and statistical analysis in result has been adjusted for better explanation. Please see the attachment “Manuscript with track changes.docx”

6. Review comment to the author

Reviewer #2

1. The manuscript lacks novelty and more than 80% of the results are already established by previous studies. Authors tried to describe the elevated GSH activity as a novel property but the discussion section is incomplete regarding its consequences and significance with the present study.

Response/ Action: Thanks for reviewer’s great comment and suggestions as well. Though many works have conducted on prodigiosin from Serratia marcescens throughout the world, this is the first ever reported work investigated in Bangladesh. Moreover, Serratia marcescens BRL41 is the pioneer bacteria isolated from the ancient soil of Bengal basin in the Northern East region of Bangladesh. Along with prodigiosin the identification of other 9 biosynthetic gene clusters are new findings here as most of the studies have focused on only identification of prodigiosin biosynthetic cluster. Additionally, we have incorporated a new finding regarding the presence of virulence and resistance genes in BRL41, a data reported for the first time. To the best of our knowledge, determination of cellular GSH is our novel finding as no other study has conducted yet. 

Based on reviewer’s comment we have added information about the consequence and significance of GSH measurement in our discussion section. Please go though the revised manuscript.

Before revision: Based on GHS level measurement the study suggests prodigiosin ability to promote glutathione biosynthetic pathway though further investigations are required to verify the role of extract on glutathione synthetase enzyme. This is the first ever reported data describing the role of prodigiosin on facilitating GHS synthesis as an antioxidant

After revision: As a signaling molecule, GSH is involved in antioxidant defense protecting cell from oxidative damage by Reactive Oxygen Species (ROS). Over the range of 125-500 µg/ml, prodigiosin significantly increased (p< 0.05) GSH in treated cell at concentration 250 and 500 µg/ml “Fig 8” where 125 µg/mL failed to stimulate GSH synthesis. This finding is an evidence towards prodigiosin’s efficacy as an antioxidation stimulating agent.

Before revision: AntiSMASH analysis for the secondary metabolite producing BGC showed the presence of a prodigiosin producing gene cluster in BRL41 sequence data. Among the seven gene clusters found in BRL41, the hybrid cluster “Prodigiosin NRPS” showed 100% similarity with the query sequence (Table 1). Furthermore, pigC, pigH, pigI, and pigJ were the core biosynthetic genes responsible for prodigiosin production, along with five other additional biosynthetic genes. The regulatory gene of this cluster was cueR “Fig 2”

Table 1: Biosynthetic gene clusters for secondary metabolite producing genes in BRL41 strain using AntiSMASH

BGS Type Region Most similar known Cluster Similarity

Prodigiosin, NRPS 446243- 507432 Prodigiosin 100%

nNRPS 43621- 91569 microcin H47 20%

NRPS 1-48188 Vulnibactin 18%

Thiopeptide 87897-114341 O-antigen 14%

RRE-containing 1-1641 lankacidin C 13%

NRPS 382752-459267 Ravidomycin 5%

NRPS 67669-111604 Xantholipin 4%

After revision: AntiSMASH analysis for the secondary metabolite producing BGC visualized the presence of 10 different biosynthetic gene clusters in BRL41 chromosome where the hybrid cluster “Prodigiosin NRPS” showed 100% similarity with the query sequence (Table 1). The core prodigiosin biosynthetic genes pigC, pigH, pigI, and pigJ co-existed with five additional biosynthetic genes and all of them were regulated by cueR “Fig 2”. Similarly, two other clusters, supposed to produce ririwpeptide and yersinopine, exhibited 100% similarity with query sequences in BGC analysis.

Table 1: Biosynthetic gene clusters for secondary metabolite producing genes in BRL41 strain using AntiSMASH

BGC Type From To Most similar known cluster Similarity

prodigiosin,NRPS 446,225 507,414 Prodigiosin 100%

NRPS 132,409 176,344 ririwpeptide A/ririwpeptide B/ririwpeptide C 100%

opine-like-metallophore 121,810 143,907 Yersinopine 100%

NRP-metallophore,NRPS 304,172 354,869 trichrysobactin/cyclic trichrysobactin/chrysobactin/dichrysobactin 46%

NRPS 1,076,754 1,125,264 trichrysobactin/cyclic trichrysobactin/chrysobactin/dichrysobactin 38%

NRPS 382,689 459,204 5-dimethylallylindole-3-acetonitrile 33%

NRPS 7,283 45,973 microcin E492 18%

Thiopeptide 136,026 162,470 O-antigen 14%

RRE-containing 1 1,641 lankacidin C 13%

Betalactone 119,114 144,783 

Presence of antibiotic resistance and virulence genes

In “perfect” algorithm setting of CARD database, no resistance gene was found in the genome of the isolate. On the other hand, with very low percentage of identity, some resistance genes were found in “Strict” setting but they were against commonly available beta-lactam, first-generation antibiotics. Though the VFDB analysis showed 25 virulence genes in the chromosome, they were not pathogenic genes at all cause being a gram-negative bacterium, having some structural endotoxins in Serratia is quite normal.

2. Authors also claimed in the response to reviewer section that prodigiosin affects the membrane integrity in gram positive and gram-negative bacteria, which is the first report; but effect of prodigiosin on cell membrane integrity has been described previously through fluorescence microscopy and ion leakage assay.

 Response/action: Although few researches have been reported regarding membrane integrity assay through fluorescence microscopy and ion leakage assay, most of them were measured the released nucleic acid concentration whereas we have measured the protein concentration as indirect symbol of membrane disruption upon treatment with prodigiosin

3. Authors mentioned WGS of the bacterial isolate, but did not provided the access to those results. As per the journal policy, any molecular data generated should be deposited to public database and accession numbers to be mentioned in the manuscript.

Response/ action: According to the requirement of journal WGC data generated from this study has deposited at the NCBI sequence read archive (SRA) under accession Bio project PRJNA998550.

4. Additionally, authors tried improved the English language use and sentence writing skills in the revised version, however there are still some major issues with grammars and sentence construction, which need to be checked properly.

Response/ action: According to reviewer’s suggestion we have upgraded our writing and checked the sentences properly so that it can reach to a standard form. Please check the revised copy of manuscript. Please go through the documents “Manuscript with track changes.docx”

Reviewer#3

1. The selection of the strain still needs some clarifications. What they did was based on the screening of pigmented strains or search for Serratia species. The sentence in line 99-100 needs more clarification

Response/Action: Thank you for reviewer’s nice suggestion. According to reviewer’s demand we have tried to clarify the fact in line 99-100 that the selection of Serratia marcescens was based on red pigmented strain which was further confirmed as serratia.

Before revision: Smi 41 was screened out from the mixed microbiota of soil through a sequential screening at varying temperature and incubation time, that resulted in appearance of few red colonies on nutrient agar medium.

After revision: In search of a pigment producing isolate, BRL41 was screened out from the mixed microbiota of soil through a sequential screening at varying temperature and incubation time that resulted in appearance of few colored colonies on nutrient agar medium. Some of the pigmented colonies were picked up and sub-cultured several times to obtain pure culture. From there, we dealt with only red pigmented one and the morphological trait was determined by microscopic observation and biochemical analysis in a semi-automated micro station ID system using the software program MicrologTM 6.2 according to the manufacturer's instructions. For further confirmation whole genome sequencing was performed using Illumina MiniSeq platform.

2. References should be updated (use articles published on 2021, 2022 and 2023). there are many articles recently published focusing on prodiginines production and characterization.

Response/Action: Thank you for reviewer’s suggestion. We have replaced some of the references with updated one as well as cited many recent works in our manuscript that were not mentioned in previous copy. Almost 80% of citation here are from the paper published between year 2015 to 2023. Please have a look on the reference section of the manuscript.

3. Change of color with pH, FTIR and UV are considered being sufficient to conclude about the structure of the compound? I think authors have to confirm with MS or NMR analysis

Response/Action: Thank you for the comment reviewer has made here. In our study we worked with crude extract of prodigiosin and based on the alignment of our results with previous findings regarding pH, UV, TLC, FTIR, we have simply reported our metabolite as prodigiosin rather than concluding its structure. To the best of our knowledge, prodigiosin is the only red pigmented metabolites synthesized by Serratia marcescens which is another support towards our result. For your kind consideration, I would like to mention that right now it wouldn’t be possible for us to analyze the compound with MS or NMR due to some unavoidable reasons. We hope that in future study we will definitely consider you suggestion.

4. The mechanism of action still needs further investigations to be confirmed

Response/Action: Thank you so much for reviewer’s suggestion. The assay here served as an indirect method to assess bacterial membrane impairment, one of the possible mechanisms of action of prodigiosin in bacterial cells. Therefore, the increase of protein concentrations of prodigiosin exposed culture over time is a support towards membrane disruption that resulted in release of protein onto medium. In future we will study its underlying mechanism in disruption of lipid bilayer as well as the other targeted cellular component with which prodigiosin may interact.

Minor comments:

Response/Action: Thank you for the comment reviewer has made here. We have checked our manuscript and tried to improve English up to a standard to make it fit for publication on “PLOS ONE”.

-According to reviewer’s suggestion we have changed the italic form of ‘sp’with ‘sp’ throughout the manuscript.

- In the calculation of prodigiosin units/cell, a factor of 1000 is included in the formula to avoid working with small numbers (<1) according to the process described by (Yip et al. 2021)

-For the analysis of prodigiosin in TLC various solvents like methanol, DMSO, ethyl acetate, acetonitrile is used commonly. We herein have chosen ethyl acetate as it showed a greater movement of both mobile phase and compound from its application point. Therefore, use of this solvent in further purification steps can lead to a better separation of prodigiosin from its closely related impurities.

- Among 4 steps in formation of bacterial biofilm, adhesion is considered as the first and most crucial steps. These adhesion leads to biofilm formation that imposes serious challenge in healthcare- associated infections, particularly those involving medical devices implementation. Therefore, the current researches in the medical field are oriented towards bacterial adhesion prevention rather than biofilm eradication. By taking this issue in consideration, we have conducted only the anti- adhesive study here.

- In antimicrobial activity determination section of our experiment we chose chloramphenicol 30 µg disc as control because chloramphenicol is a broad spectrum antibiotic active against both Gram-positive and Gram-negative pathogens. Additionally, based on the comment of reviewer on comparative analysis of chloramphenicol and prodigiosin, we have presented the data in different format in “table 2” to correlate the size of inhibition zone with different bacterial isolates.

 Before revision:

 Concentration Tested bacteria

 (µg/ml) E. coli Staphylococcus aureus Listeria monocytogens ATCC-3193 Pseudomonas aeruginosa ATCC- 9027 Salmonella enterica ATCC-10708

Prodigiosin 500 25.61 ± 0.15a 18.66 ± 0.0.33b 18.56 ± 0.22b 24.73 ± 0.15b 19.60 ± 0.05b

 250 16.89 ± 0.40b 14.00 ± 0.45c 13.00 ± 0.20c 16.61 ± 0.11c 7.77 ± 0.39b

 125 No zone 9.00 ± 0.00d 9.00 ± 0.00d No zone No zone

 +ve (control) 26.00 ± 0.58a 27.67 ± 0.33a 27.67 ± 0.33a 26.33 ± 0.33a 25.00 ± 0.00a

Values are expressed as Mean ± SEM (n=3) where SEM signifies standard error of mean. Analysis was performed with one-way ANOVA followed by Tukey Post Hoc comparisons. Mean containing different letters in same column describe significant difference of result at 5% level of Significance (p≤0.05)

After revision:

 Concentration Tested bacteria

 (µg/ml) E. coli Staphylococcus aureus Listeria monocytogens ATCC-3193 Pseudomonas aeruginosa ATCC- 9027 Salmonella enterica ATCC-10708

Prodigiosin 500 25.61 ± 0.15a 18.66 ± 0.0.33b 18.56 ± 0.22b 24.73 ± 0.15a 19.60 ± 0.05b

 250 16.89 ± 0.40a 14.00 ± 0.45b 13.00 ± 0.20b 16.61 ± 0.11a 7.77 ± 0.39c

 125 No zone 9.00 ± 0.00d 9.00 ± 0.00d No zone No zone

 (control) 26.00 ± 0.58b 27.67 ± 0.33a 27.67 ± 0.33a 26.33 ± 033a, b 25.00 ± 0.00b

Values are expressed as Mean ± SEM (n=3) where SEM signifies standard error of mean. Analysis was performed with one-way ANOVA followed by Tukey Post Hoc comparisons. Mean containing different letters in same row describe significant difference of result at 5% level of Significance (p≤0.05)

- A reference has been added to the quantification equation of prodigiosin to support it

- Thank you for reviewer’s comment about the claim that prodigiosin is mostly produced by non-pathogenic stain of Serratia marcescens. For better understanding and clarification of this point we have rearranged the sentence.

Before revision: Researchers claim that prodigiosin is produced mostly by the non-pathogenic strain of Serratia marcescens(14).

After revision: Though the both pigmented and non-pigmented variants of Serratia marcescens are pathogenic for human, researchers claim that the non-pigmented strains are more virulent due to cytotoxin production and antibiotic resistance.

Reviewer#4

1. The manuscript describes the In silco detection of prodigiosin biosynthesis gene cluster in Serratia sp. BRL41 and, also describes the isolation and antimicrobial activity of prodigiosin against gram- positive and gram- negative bacteria. However, most of the results presented here were previously described by many researchers. Hence, novelty of this study is missing. This study also possesses some experimental drawbacks in the study design as well as discussion of the results. 

Response: Thanks for reviewer’s great comment and suggestions as well. Though many works have conducted on prodigiosin of Serratia marcescens throughout the world, this is the first ever reported work investigated in Bangladesh. Moreover, Serratia marcescens BRL41 is the pioneer bacteria isolated from the ancient soil of Bengal basin in the Northern East region of Bangladesh. Detection of 10 biosynthetic gene clusters is a new finding here as most of other studies have conducted only in identification of prodigiosin gene cluster. Additionally, we have incorporated a new finding regarding the presence of virulence and resistance genes in BRL41, a data reported for the first time. Determination of cellular GSH is our novel finding as no other study has conducted yet. Based on reviewer’s comment we have added information about the consequence and significance of GSH measurement in our discussion section. Please see the response and action section of comment 1 of reviewer#2

2. The authors mentioned WGS analysis of isolate BRL41 but, results and methodologies were not described in detail. Link for the WGS data is also provided. The phylogenetic analysis was performed but the accession no for BRL41 was not provided. The authors failed to describe the specialty of this isolate as the BGC of prodigiosin is similar to those reported previously

Response: Thanks to reviewer for this comment. Following the instruction of reviewer, the methodology and result sections have been rewritten. Besides, accession number and link for WGS has also added in the manuscript. In our study along with Prodigiosin, BGC analysis of genome showed the presence of other secondary metabolite synthesizing gene clusters that have opened a new door of possibilities to avail this isolate for the production of some important bioproducts. Additionally, absence of virulence and resistance gene is genome of BRL41 verifies its compatibility as a strain that can be used commercially. 

 Action: In material and method Section: DNA from the isolate was extracted using Wizard® Genomic DNA Purification Kit (Promega, catalog no. PR-A1120) and whole genome sequencing was done by Illumina MiniSeq platform. Library preparation was conducted as per the manufacturer’s protocol. After obtaining the raw data, the assembly of the genome was performed using the Shovill pipeline (21) that utilized SPAdes as its core and is specific for bacterial genomes from Illumina paired end reads. Raw sequencing data generated from this study are deposited at the NCBI sequence read archive (SRA) under accession Bio project PRJNA998550. Phylogenetic analysis was performed by Type (Strain) Genome Server (TYGS) using maximum-likelihood method (22)and the phylogenetic tree was visualized using iTOL(23). For further confirmation of the similarity of the isolate, average nucleotide identity (ANI) was calculated using FastANI (24). Biosynthetic gene clusters responsible for the secondary metabolite production were identified using antiSMASH (25). Moreover, antibiotic resistant genes and virulence genes were profiled by CARD browse (26) and VFDB analyzer. This identified bacterial isolate was used to conduct further studies.

In result section: Based on the utilization of different metabolic substrates and the ability to grow in the presence of certain inhibitory chemicals, the result from MicroStation GEN III database 2.8.0 library showed 88% similarity of this strain with Serratia marcescens. Genome sequencing data depicted the highest sequence similarities with Serratia marcescens that was further supported by ANI results, showing 98.15% nucleotide similarity with the reference genome of Serratia marcescens (GCF_003516165.1) Sequence alignment and construction of the phylogenetic tree revealed that the isolate is very closely related to strain Serratia marcescens subsp., sakeunsis KCTC 42172, and Serratia marcescens ATCC 13880 “Fig 1”. 

3. The discussion section should emphasise on the scope for the use of prodigiosin against the target pathogens and their target mechanisms at molecular level. Additionally, the whole manuscript should be properly checked by a native English speaker for improvement of the language used.

Response/ action: Following the instructions of reviewer’s, we have reconstructed our discussion section by giving emphasis on the scope of using prodigiosin and the targeted mechanisms underlying its antimicrobial activity. Additionally, we have tried our best to improve language throughout the manuscript. Hope you will be kind enough to consider this issue as we are not native speaker and will give us the opportunity to be with “PLOS ONE”.

Some specific comments-

1.Unit should be synchronised for e.g. The gap between the value and the unit should be maintained.

2. Line 5- should mention properly the type of soil from which the bacteria were isolated.

3. Line 39- space needed in 10µg/mL (5).

4. Line 55- space needed in marcescens (14).

5. Line 57- needed a dot in etc

6. Line 58- space needed in (15) Physicochemical.

7. Line 60- space needed so check the line.

8. Line 65- the scientific name should be in italics, check the line.

9. Line 109-110- Modify the line, reduce the use of the word ‘using’ in to one time.

10. Line 110- space needed in iTOL(22), check the line.

11. Line 124-should replace the word ‘with’ by the word ‘by’.

12. Line 149- the symbol comma is not needed after the word pH.

13. Line 162- the symbol comma in ‘cool, dry’ should be replaced by the word ‘and’.

14. Line 213- Space is needed in (26) Bovine.

15. Line 331-333- Modify the language (Therefore,…..animals).

Reply/Action:

1. The gap between the value and the unit has synchronized

2. Soil type from where the bacteria was isolated has mentioned in line 5

3. Line space has added in 10 µg/mL

4. Space has added in Serratia marcescens (14)

5. A dot has included in etc

6. Space has added in (15) physicochemical

7. The line has checked and corrected accordingly

8. The name of Staphylococcus aureus has changed to italic

9. The line has modified

10. Space had added here

11. The word in the sentence “acidified with 1N NaOH” has replaced by “acidified by 1N NaOH”

12. There is no comma after pH in this line

13. The has corrected

14. A space has added in (26) and Bovine

15. We have corrected this sentence by replacing it with new one

---

## [Decision Letter · Decision Letter 2]

19 Sep 2023

PONE-D-23-01982R2In silico exploration of Serratia sp. BRL41 genome for detecting prodigiosin biosynthetic gene cluster (BGC) and in vitro antimicrobial activity assessment of secreted prodigiosinPLOS ONE

Dear Dr. Boby,

Thank you for submitting your manuscript to PLOS ONE. After careful consideration, we feel that it has merit but does not fully meet PLOS ONE’s publication criteria as it currently stands. Therefore, we invite you to submit a revised version of the manuscript that addresses the points raised during the review process.

We look forward to receiving your revised manuscript.

Kind regards,

Marcos Pileggi, Ph.D

Academic Editor

PLOS ONE

Journal Requirements:

**Additional Editor Comments:**

The authors have successfully addressed the majority of the concerns raised by the reviewers. However, in order for the manuscript to meet the criteria for publication, minor modifications need to be acknowledged and made by the authors.

Reviewers' comments:

Reviewer's Responses to Questions

**Comments to the Author**

1. If the authors have adequately addressed your comments raised in a previous round of review and you feel that this manuscript is now acceptable for publication, you may indicate that here to bypass the “Comments to the Author” section, enter your conflict of interest statement in the “Confidential to Editor” section, and submit your "Accept" recommendation.

Reviewer #2: All comments have been addressed

Reviewer #3: All comments have been addressed

2. Is the manuscript technically sound, and do the data support the conclusions?

Reviewer #2: Yes

Reviewer #3: Yes

3. Has the statistical analysis been performed appropriately and rigorously? 

Reviewer #2: Yes

Reviewer #3: Yes

4. Have the authors made all data underlying the findings in their manuscript fully available?

Reviewer #2: Yes

Reviewer #3: Yes

5. Is the manuscript presented in an intelligible fashion and written in standard English?

Reviewer #2: Yes

Reviewer #3: Yes

6. Review Comments to the Author

Reviewer #2: 1. The manuscript has been significantly improved. However, the membrane integrity part still suffers from lack of proper justification. Protein leakage assay is acceptable but it can not justify the loss of membrane integrity, unless it is linked to direct evidences of membrane damage. Authors should link their findings to those of previous studies to justify their results. Please refer to the findings of Lependa et al. (2015), Kimyon et al. (2016), Suryawanshi et al. (2017), Hazarika et al. (2021), and Ravindran et al. (2020).

2. All the references need to be synchronized, especially those having an internet link.

Reviewer #3: Authors have taken into considerations all my comments in the revised manuscript. I have just noticed a bad resolution of figures. It would be better to improve during the publication process.

7. PLOS authors have the option to publish the peer review history of their article (what does this mean?). If published, this will include your full peer review and any attached files.

Reviewer #2: No

Reviewer #3: **Yes: **Sami Mnif

---

## [Author Response · Author response to Decision Letter 2]

2 Oct 2023

All the comments of reviewers have addressed and the file is attached as "Responses to reviewers

---

## [Decision Letter · Decision Letter 3]

25 Oct 2023

In silico exploration of Serratia sp. BRL41 genome for detecting prodigiosin biosynthetic gene cluster (BGC) and in vitro antimicrobial activity assessment of secreted prodigiosin

PONE-D-23-01982R3

Dear Dr. Boby,

We’re pleased to inform you that your manuscript has been judged scientifically suitable for publication and will be formally accepted for publication once it meets all outstanding technical requirements.

Kind regards,

Marcos Pileggi, Ph.D

Academic Editor

PLOS ONE

Additional Editor Comments (optional):

Reviewers' comments:

Reviewer's Responses to Questions

**Comments to the Author**

1. If the authors have adequately addressed your comments raised in a previous round of review and you feel that this manuscript is now acceptable for publication, you may indicate that here to bypass the “Comments to the Author” section, enter your conflict of interest statement in the “Confidential to Editor” section, and submit your "Accept" recommendation.

Reviewer #2: All comments have been addressed

2. Is the manuscript technically sound, and do the data support the conclusions?

Reviewer #2: Yes

3. Has the statistical analysis been performed appropriately and rigorously? 

Reviewer #2: Yes

4. Have the authors made all data underlying the findings in their manuscript fully available?

Reviewer #2: Yes

5. Is the manuscript presented in an intelligible fashion and written in standard English?

Reviewer #2: Yes

6. Review Comments to the Author

Reviewer #2: My concerns have been addressed. The manuscript can be accepted for publication in this format. I congratulate the authors for successful revision of the manuscript.

7. PLOS authors have the option to publish the peer review history of their article (what does this mean?). If published, this will include your full peer review and any attached files.

Reviewer #2: No

---

## [Editor Report · Acceptance letter]

7 Nov 2023

PONE-D-23-01982R3 

In silico exploration of *Serratia* sp. BRL41 genome for detecting prodigiosin biosynthetic gene cluster (BGC) and in vitro antimicrobial activity assessment of secreted prodigiosin 

Dear Dr. Boby:

I'm pleased to inform you that your manuscript has been deemed suitable for publication in PLOS ONE. Congratulations! Your manuscript is now with our production department. 

Kind regards, 

on behalf of

Dr. Marcos Pileggi 

Academic Editor

PLOS ONE